# Tactile modulation of memory and anxiety requires dentate granule cells along the dorsoventral axis

Chi Wang [1,2,8], Hui Liu [1,2,8], Kun Li[1,8], Zhen-Zhen Wu[1], Chen Wu[1,2], Jing-Ying Yu[1], Qian Gong[1], Ping Fang[1], Xing-Xing Wang[3], Shu-Min Duan [2], Hao Wang [2], Yan Gu [4], Ji Hu[5], Bing-Xing Pan [6], Mathias V. Schmidt [7], Yi-Jun Liu[1] & Xiao-Dong Wang [1,2✉]

Touch can positively influence cognition and emotion, but the underlying mechanisms remain unclear. Here, we report that tactile experience enrichment improves memory and alleviates anxiety by remodeling neurons along the dorsoventral axis of the dentate gyrus (DG) in adult mice. Tactile enrichment induces differential activation and structural modification of neurons in the dorsal and ventral DG, and increases the presynaptic input from the lateral entorhinal cortex (LEC), which is reciprocally connected with the primary somatosensory cortex (S1), to tactile experience-activated DG neurons. Chemogenetic activation of tactile experience-tagged dorsal and ventral DG neurons enhances memory and reduces anxiety respectively, whereas inactivation of these neurons or S1-innervated LEC neurons abolishes the beneficial effects of tactile enrichment. Moreover, adulthood tactile enrichment attenuates early-life stress-induced memory deficits and anxiety-related behavior. Our findings demonstrate that enriched tactile experience retunes the pathway from S1 to DG and enhances DG neuronal plasticity to modulate cognition and emotion.

[1] Department of Neurobiology and Department of Psychiatry of Sir Run Run Shaw Hospital, Zhejiang University School of Medicine, 310058 Hangzhou, China.
[2] NHC and CAMS Key Laboratory of Medical Neurobiology, MOE Frontier Science Center for Brain Research and Brain-Machine Integration, School of Brain Science and Brain Medicine, Zhejiang University, 310058 Hangzhou, China. [3] Department of Anesthesiology, Technische Universität München/Klinikum Rechts der Isar, 81675 Munich, Germany. [4] Center of Stem Cell and Regenerative Medicine, and Department of Neurology of the Second Affiliated Hospital, Zhejiang University School of Medicine, 310058 Hangzhou, China. [5] School of Life Science and Technology, ShanghaiTech University, 201210 Shanghai, China. [6] Laboratory of Fear and Anxiety Disorders, Institute of Life Science, Nanchang University, 330031 Nanchang, China. [7] Max Planck Institute of Psychiatry, 80804 Munich, Germany. [8] These authors contributed equally: Chi Wang, Hui Liu, Kun Li. ✉email: xiaodongwang@zju.edu.cn

The sense of touch is fundamental for object recognition, motor control, and social communication in different mammalian species, including rodents and humans. Touch not only conveys discriminative and affective information[1], but also can modulate cognition and emotional responses[2,3]. For example, skin-to-skin care of preterm infants improves cognitive development[4], while a brief parental touch alleviates social anxiety in children[5]. Moreover, neonatal tactile enrichment reduces anxiety levels and enhances working memory in adult rats[6,7]. Conversely, impaired tactile processing is implicated in cognitive and anxiety disorders[8–10]. Although the modulatory effects of touch on cognition and emotion have been documented, the underlying circuit and cellular mechanisms remain elusive.

Tactile information is conveyed and processed via serial and parallel pathways. Innocuous mechanical stimuli such as pressure and vibration are encoded by cutaneous low-threshold mechanoreceptors and transmitted via ascending somatosensory pathways to the primary somatosensory cortex (S1), where tactile signals are decoded and processed[11,12]. Tactile inputs are further processed and integrated together with inputs from other sensory modalities in higher order association cortices[13,14]. As a component of the parahippocampal region, the lateral entorhinal cortex (LEC) is proposed to gate the flow of uni- and multimodal sensory information to the hippocampus[15,16], which supports declarative memory by relational processing of spatial and nonspatial information[17]. Emerging evidence indicates that the LEC contributes to somatosensory processing[18]. The LEC is reciprocally connected with the S1[19–21], and its layer 2 reelin-expressing neurons relay processed sensory information to the dentate gyrus (DG)[22]. Notably, dentate granule cell activity can be altered by tactile stimulation[23] or deprivation[24]. Since memory and anxiety are differentially modulated by DG neurons along the hippocampal dorsoventral axis[25,26], it is possible that tactile inputs converge on the DG via the LEC to shape cognitive and emotional processes.

Here we used a novel mouse model of unimodal tactile enrichment and investigated whether and how DG neurons modulated the influence of enriched tactile experience on memory and anxiety. To dissect the potentially differential roles of the dorsal DG (dDG) and the ventral DG (vDG) in tactile enrichment-evoked cognitive and emotional effects, we combined functional labeling of neurons[27] and chemogenetics to manipulate tactile experience enrichment-activated neuronal ensembles ("TEE-tagged" neurons) in dDG or vDG. The involvement of LEC neurons that link S1 and TEE-tagged DG neurons in tactile enrichment effects was also explored. Moreover, we examined whether tactile enrichment in adulthood could ameliorate the adverse impact of early postnatal stress. Our results show that recurrent tactile enrichment enhanced synaptic plasticity in dDG and vDG granule cells to improve memory and reduce anxiety, respectively, and attenuated early life stress-induced behavioral and structural abnormalities.

## Results

**Multimodal enrichment increased neuronal activation in DG and S1.** Environmental enrichment provides augmented sensory, motor, and social stimuli (Fig. 1a and Supplementary Movie 1), and positively modulates neural plasticity and behavior[3,28]. We quantified c-fos$^+$ activated neurons to evaluate the response of dentate gyrus and primary sensory cortices to such multimodal enrichment (Fig. 1b and Supplementary Fig. 1a–f). After multimodal enrichment, the number of activated granule cells in dDG and vDG and activated neurons in specific layers of S1, piriform cortex (Pir), primary visual cortex (V1), and primary auditory

cortex (Au1) was increased (Fig. 1c, d). Notably, S1 layer 4 that receives tactile information from the thalamus showed more prominent activation than layer 4 of V1 and Au1 (Fig. 1e). These data suggest tactile input as an important sensory source under multimodal enrichment, and raise the possibility of S1-associated neural circuits modulating the enrichment effects on cognition and emotion.

**Tactile enrichment improves memory and alleviates anxiety.** To better differentiate between tactile and other forms of enrichment, we used a mouse model of unimodal tactile enrichment (Fig. 1f, Supplementary Fig. 2a–j and Supplementary Movie 2) to examine the cognitive and emotional impact of enriched tactile experience. Compared to controls, mice that received recurrent tactile enrichment performed better in the novel object recognition task that requires the perirhinal cortex and the hippocampus[29] (Fig. 1g) and the object location task that depends on the hippocampus[30] (Fig. 1h). In the temporal order and object-in-place tasks that require multiple cortical regions[31], no difference was noticed between groups (Supplementary Fig. 3a, b). Mice with tactile enrichment also showed reduced anxiety-related behavior, shown by their spending more time in the open field center (Fig. 1i) and open arms of the elevated plus maze (Fig. 1j) as well as more head dips (Supplementary Fig. 3c) than control mice. In the light-dark box test, the two groups performed similarly (Supplementary Fig. 3d).

**Tactile enrichment modulates neuronal activation in DG and S1.** Considering the positive effects of tactile enrichment on memory and anxiety, we examined the activation of dentate granule cells by tactile enrichment and compared the effects of different enrichment conditions (Fig. 2a, b). Repeated tactile enrichment did not significantly alter the density of c-fos$^+$ activated cells in dDG, but enhanced neuronal activation in vDG (Fig. 2c). Increasing the rows of bead curtain in the tactile enrichment cage could not further augment DG neuron activation (Supplementary Fig. 3e, f). As revealed by nearest neighbor distances (Fig. 2d), a measure of spatial point pattern[32], activated cells in dDG and vDG appeared more clustered in the tactile enrichment group (median: 0.0247 and 0.0264 mm respectively) compared to the controls (median: 0.0274 and 0.0356 mm). By contrast, multimodal enrichment increased neuronal activation and rearranged activation pattern in both dDG and vDG. In addition, although socially housed mice and singly housed controls differed in spatial patterns of activated DG neurons, both groups performed similarly in the object location and elevated plus maze tasks (Supplementary Fig. 3g, h).

We further compared the effects of tactile enrichment, multimodal enrichment and social housing on neuronal activation in S1 subregions (Supplementary Fig. 4a–i). After tactile enrichment, the density of c-fos$^+$ neurons in the barrel field and trunk regions of S1 was significantly increased. Multimodal enrichment exerted similar effects in these areas and increased activated neuron density in deep layers of S1 forelimb, hindlimb, and shoulder regions. In comparison, the modulation of neuronal activation by social housing was limited to layers 2-4 of the barrel field. These data reveal that tactile enrichment differentially activates dDG and vDG neurons and increases neuronal activation in S1.

**Tactile enrichment modulates behavior and neuronal activity in a temporally specific manner.** By varying the duration of tactile enrichment and the interval between tactile enrichment and behavioral testing, we characterized how tactile experience

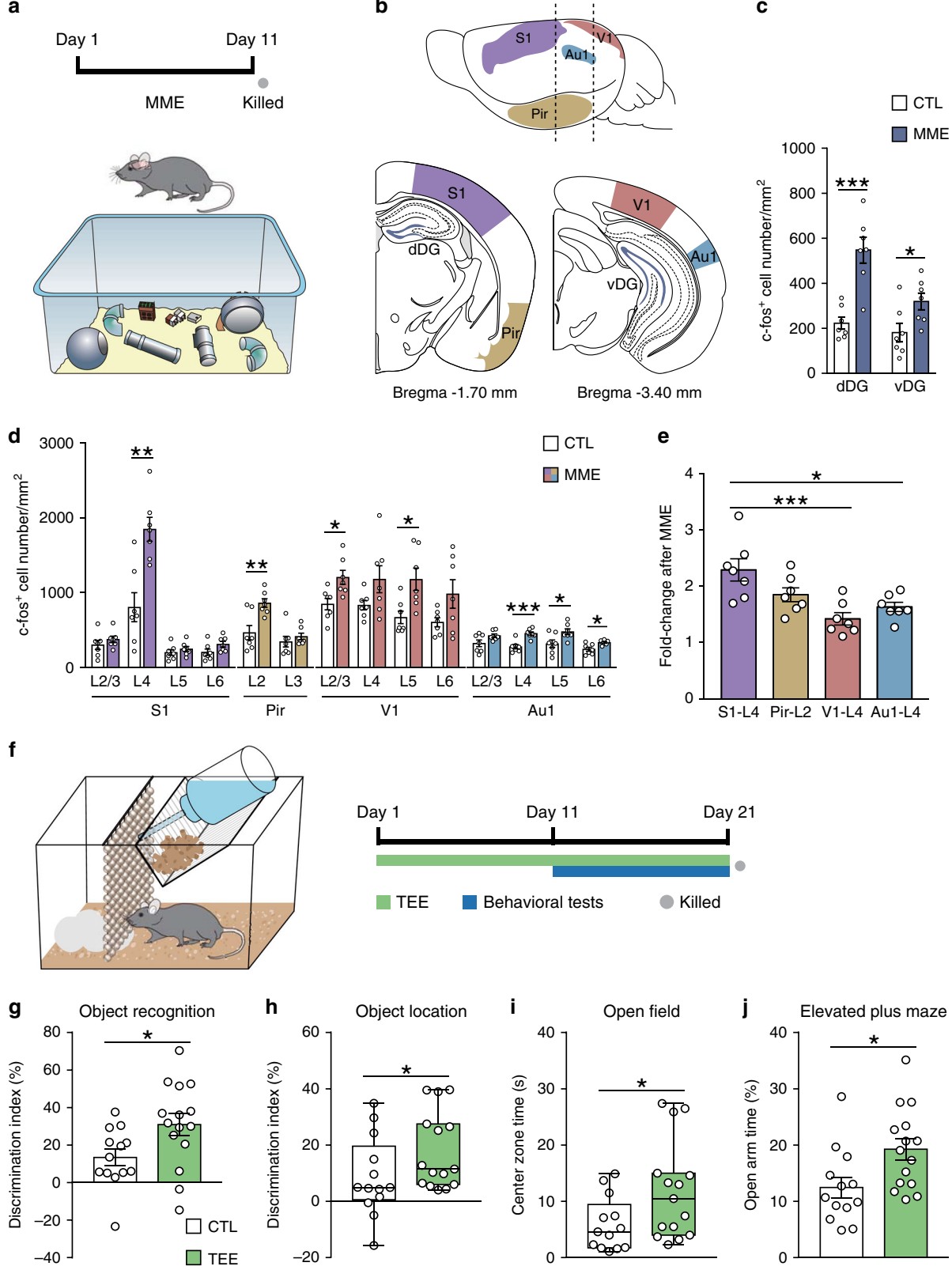

could modulate memory, anxiety and DG neuronal activation over time. Short-term exposure to enriched tactile environment for 1.5 days was insufficient to affect spatial memory or anxiety level (Fig. 3a–c). Following a 10-day tactile enrichment, positive influences on behavior were still detectable after 1 but not

4 weeks, indicating that recurrent and cumulative tactile enrichment is needed to modulate cognition and emotion.

Since the behavioral effects of repeated tactile enrichment could last for at least 1 week, we examined its temporal effects on DG neuron activation (Fig. 3d). Immediately after tactile

**Fig. 1 Activation of cortical neurons after multimodal enrichment (MME) and the behavioral effects of tactile experience enrichment (TEE). a** MME cage setup and experimental timeline. $n = 7$ mice per group. **b** Lateral and coronal views of the regions of interest. Dashed lines in the top panel indicate the levels of two coronal sections shown in bottom panels. **c** The density of activated neurons in dorsal and ventral parts of DG was increased in the MME group compared to the control (CTL) group. **d** MME activated neurons in specific layers (L) of S1, Pir, V1, and Au1. **e** S1 layer 4 neurons showed higher degree of activation than layer 4 of V1 and Au1 after MME. **f** TEE cage design and experimental timeline. Mice were singly housed in standard cages provided with a row of glass bead curtain and a piece of cotton nesting material for 20 days. During the last 10 days of TEE, multiple memory and anxiety tests were performed. **g** Compared to CTL mice ($n = 13$), TEE mice ($n = 15$) showed better novel object recognition performance. **h** TEE enhanced spatial memory performance in the object location task. **i** TEE mice spent more time in the open field center than CTL mice. **j** TEE mice spent more time in open arms of the elevated plus maze than CTL mice. For bar charts, data are presented as mean ± SEM. For box plots in (**h**) and (**i**), circles represent data points, center lines indicate medians, the lower and upper bounds of boxes correspond to the 25th and 75th percentiles respectively, and whiskers indicate minima and maxima. *$P < 0.05$, **$P < 0.01$, ***$P < 0.001$. Detailed statistics are provided in Supplementary Table 2. Source data are provided as a Source Data file.

enrichment, activated neurons in dDG and vDG were more clustered (median: 0.0304 and 0.0359 mm) compared to the controls (median: 0.0334 and 0.0487 mm), and the density of activated vDG neurons was increased (Fig. 3e–g). However, at 1 week after tactile enrichment, c-fos$^+$ neurons in dDG and vDG were again more dispersed (median: 0.0366 and 0.0487 mm). The discordance between neuronal activation and behavioral data suggests that lasting changes in activated DG neurons may contribute to the effects of tactile enrichment on memory and anxiety.

**Tactile enrichment increased mushroom spine density in dentate granule cells.** Multimodal enrichment increases adult hippocampal neurogenesis, which in turn positively modulates cognitive and emotional behavior[33,34]. We evaluated the effects of tactile enrichment on the generation and maturation of adult-born dentate granule cells. The density of minichromosome maintenance complex component 2-expressing proliferating cells and doublecortin$^+$ differentiating cells in dDG and vDG was similar between the control and tactile enrichment groups (Supplementary Fig. 5a, b). The immunoreactivity of calbindin (a marker of mature granule cells), vesicular GABA transporter (an inhibitory synapse marker), and mineralocorticoid and gluco-corticoid receptors (key modulators of stress response) was also comparable between groups (Supplementary Fig. 5c–f). More-over, tactile enrichment had no effect on the morphological maturation of retrovirus-labeled adult-born granule cells (Sup-plementary Fig. 6a–h). In line with these results, the majority of tactile enrichment-activated DG cells were calbindin$^+$ mature neurons (Supplementary Fig. 6i, j). Therefore, unlike multimodal enrichment, unimodal tactile enrichment did not modulate the proliferation, differentiation nor maturation of newly generated DG neurons.

We performed Golgi staining to examine whether tactile enrichment remodeled the dendrites and spines of dentate granule cells with mature morphology (Fig. 4a). Tactile enrich-ment did not change dendritic length nor dendritic complexity of DG neurons (Fig. 4b, c and Supplementary Fig. 7a), but subtly reduced branch point number in dDG neurons (Fig. 4d). In dDG neurons, tactile enrichment markedly increased mushroom spine density throughout the molecular layer without changing overall spine density (Fig. 4e–g). In vDG neurons, the density of mushroom spines in the medial and outer molecular layers, which receive projections from the medial entorhinal cortex and LEC respectively, was increased by tactile enrichment while total spine density was decreased. Moreover, a comparison of different enrichment conditions revealed that tactile enrichment mainly increased mushroom spine density, while multimodal enrichment markedly increased thin and total spine density in DG neurons (Supplementary Fig. 7b–i). These results indicate that tactile enrichment evokes differential dendritic spine remodeling in dDG and vDG granule cells.

**Tactile enrichment increases presynaptic input from LEC to TEE-tagged DG neurons.** We selected an adeno-associated virus (AAV) containing an activity-dependent promoter E-SARE (enhanced synaptic activity-responsive element) to drive ER$^{T2}$-CreER$^{T2}$ expression in activated neurons[27]. After systemic 4-hydroxytamoxifen (4-OHT) administration, a time window for Cre-dependent gene expression could be opened in activated neurons (Supplementary Fig. 8a–d). Using this functional labeling system, we found that the reactivation rate of dDG neurons labeled on the fifth day of tactile enrichment was higher than control neurons (Supplementary Fig. 8e–h). We further com-bined a rabies virus (RABV)-based retrograde monosynaptic tracing system (Fig. 4h, i) and identified that TEE-tagged neu-ronal ensembles in DG received inputs mainly from LEC layer 2 neurons (Fig. 4j and Supplementary Fig. 9a). In mice with enri-ched tactile experience, EGFP and DsRed co-labeled dDG and vDG neurons ("starter" cells) were innervated by significantly more LEC layer 2 neurons compared to controls (Fig. 4k, l).

Anatomical evidence from tracer injection studies shows that the LEC is reciprocally connected with the S1[19–21]. To further examine whether LEC neurons that innervate TEE-tagged DG cells receive S1 inputs, AAV-ESARE-ER$^{T2}$CreER$^{T2}$ and the retrograde RABV system were delivered to dDG or vDG, while an anterograde monosynaptic AAV2/1 that could express Cre recombinase in locally infected neurons and their first-order downstream neurons[35] was injected to the S1 of Rosa26-GFP (Ai47) reporter mice (Fig. 5a, b). Anterogradely labeled neurons that received S1 projections were observed in the LEC, the majority (~75%) of which were located in layer 5 (Fig. 5c–e and Supplementary Fig. 9b–e). Nonetheless, sparse LEC neurons that were both anterogradely and retrogradely labeled was noticed in layer 2, indicating that these neurons directly bridge S1 and DG. These results suggest that most S1 inputs are processed locally in the LEC before reaching the DG.

**The effects of tactile enrichment on memory and anxiety require dDG and vDG neurons respectively.** Based on the organization of the S1 → LEC → DG pathway, we used chemo-genetics to dissect the roles of TEE-tagged DG neurons and S1-innervated LEC neurons in the behavioral effects of tactile enrichment. We first combined functional labeling and chemo-genetics to selectively manipulate TEE-tagged neuronal ensem-bles in dDG or vDG (Fig. 6a and Supplementary Fig. 9f, g). The number of labeled TEE-tagged DG neurons was comparable among groups (Supplementary Fig. 9h, i). At 1 week after recurrent tactile enrichment when its effects still persisted, che-mogenetic activation of TEE-tagged dDG or vDG neurons did not further alter spatial memory nor anxiety-related behavior (Supplementary Fig. 10a, b). However, at 4 weeks after tactile enrichment when its impact disappeared, activation of TEE-tagged dDG and vDG neurons improved memory performance and reduced anxiety level respectively (Fig. 6b–e). Similar results

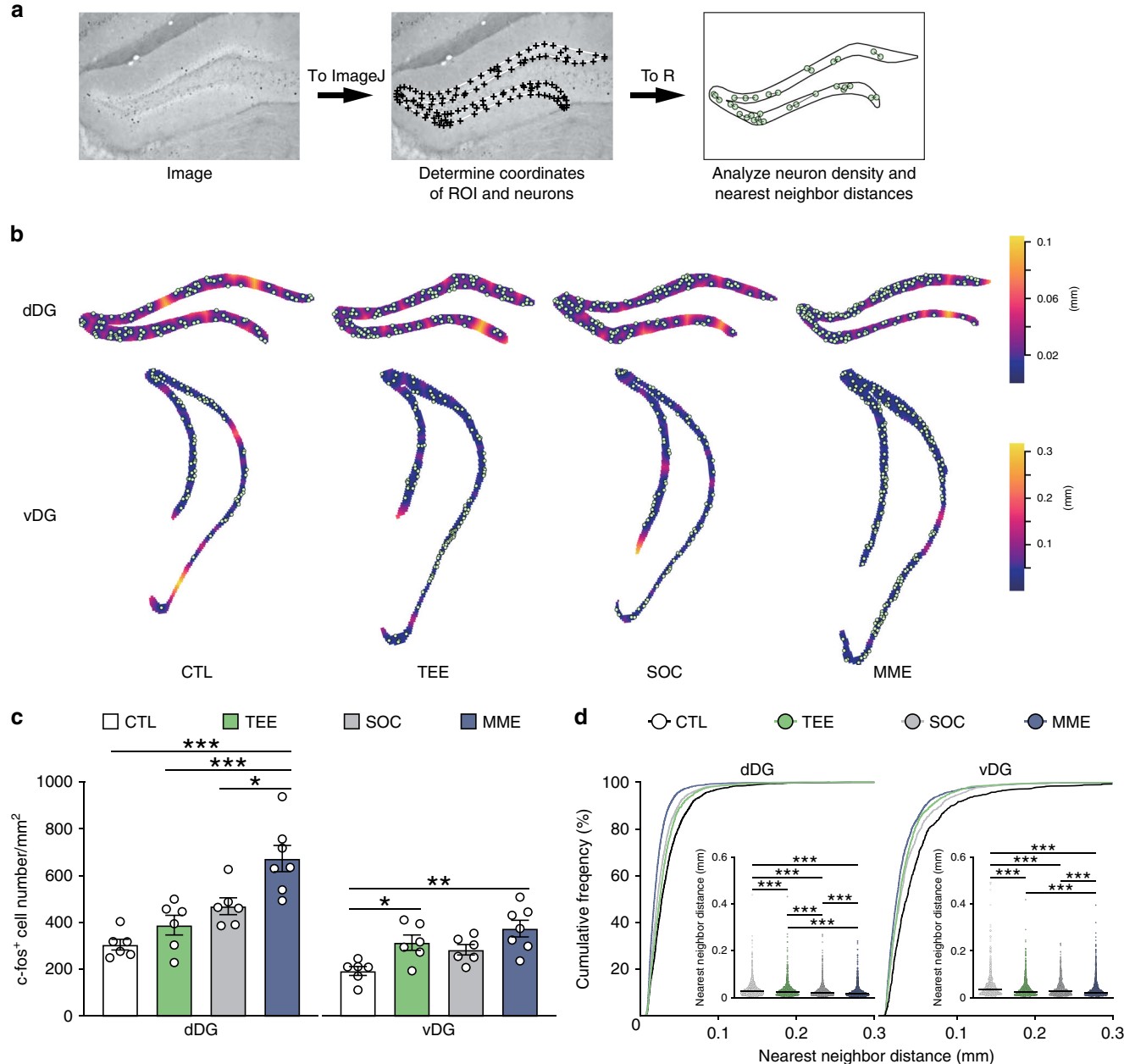

**Fig. 2 Differential modulation of DG neuron activation by tactile enrichment and multimodal enrichment. a** The pipeline for the density and spatial distribution pattern analysis of c-fos+ neurons in DG. The image was imported to ImageJ to determine the coordinates of dentate granule cell layer (ROI, shown as gray crosses connected by a white line) and c-fos+ neurons (green crosses). After importing the coordinates of ROI and neurons to R, the density and nearest neighbor distances (indicated by single- or double-headed arrows) of c-fos+ neurons were analyzed by the spatstat package. **b** Representative plots of c-fos+ neurons superimposed on empty space distance maps of dDG and vDG from the control (CTL, $n = 6$), TEE ($n = 6$), social housing (SOC, $n = 6$), and MME ($n = 7$) groups. **c** TEE for 10 days increased neuronal activation in vDG, while MME increased c-fos+ neuron density in both dDG and vDG. **d** TEE, SOC and MME altered neuronal activation pattern in dDG and vDG. $*P < 0.05$, $**P < 0.01$, $***P < 0.001$. Data in bar charts are presented as mean ± SEM. Detailed statistics are provided in Supplementary Table 2. Source data are provided as a Source Data file.

were obtained using a prolonged labeling protocol (Supplementary Fig. 10c–f).

The inactivation experiment revealed that although the object discrimination index did not differ among groups at 1 week after tactile enrichment, both control and vDG inhibition groups showed intact object recognition, whereas the dDG inhibition group failed to distinguish the relocated object from the familiar one (Fig. 6f–h), indicative of impaired spatial memory. In comparison, inactivation of TEE-tagged vDG neurons increased anxiety level without affecting memory (Fig. 6i). At 4 weeks after tactile enrichment, inactivating either

dDG or vDG neurons had no effect on memory nor anxiety (Supplementary Fig. 10g, h).

To exclude the possibility that manipulating any neuronal population in DG would affect memory and anxiety, dDG or vDG neurons that were active under home-cage conditions were labeled (Supplementary Fig. 11a, b). Chemogenetic activation of these control DG neurons failed to alter behavioral performance at 1 week or 4 weeks after labeling (Supplementary Fig. 11c–f). Taken together, these results demonstrate that TEE-activated neuronal ensembles in dDG and vDG modulate tactile enrichment effects on memory and anxiety respectively.

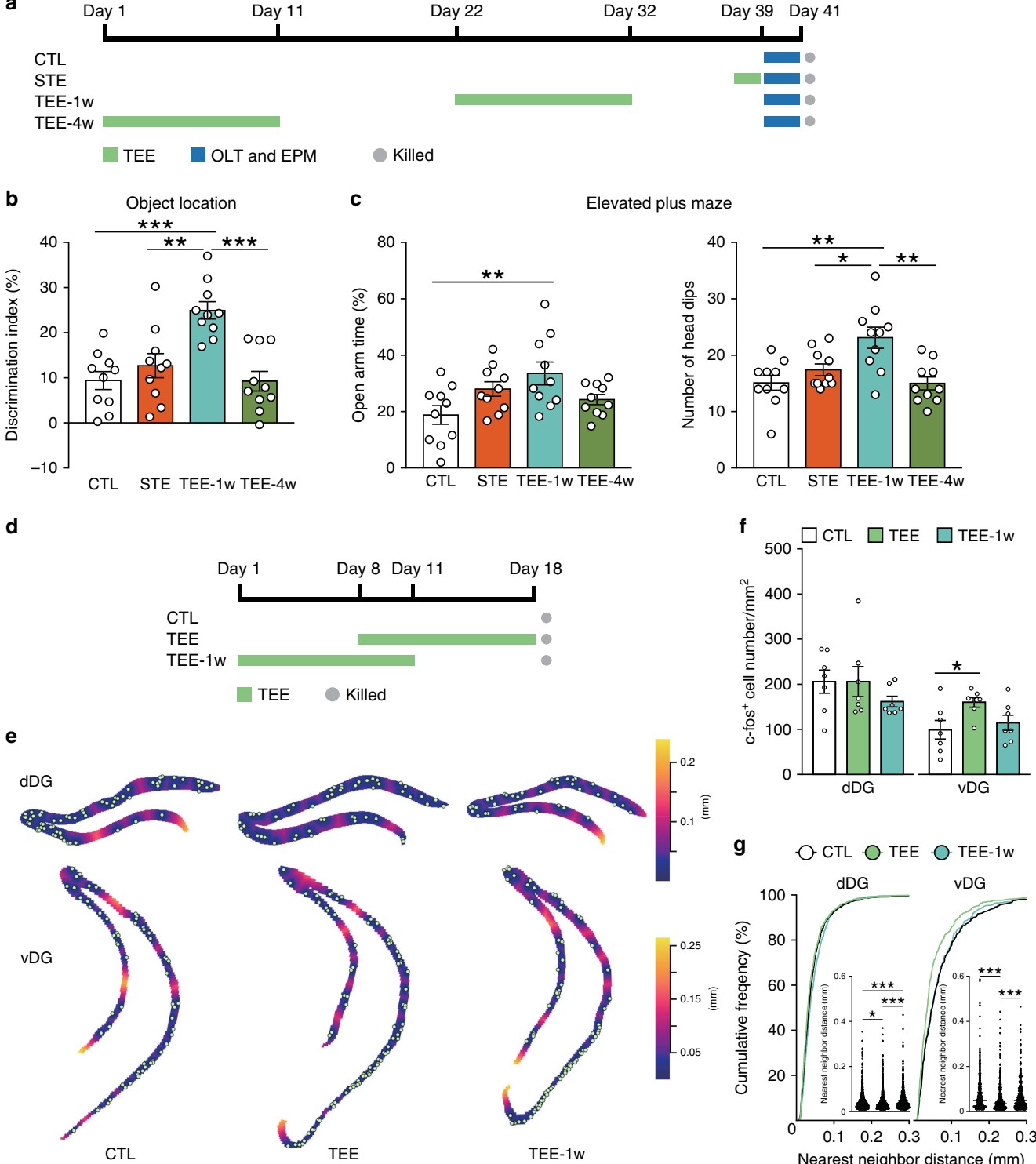

**Fig. 3 Temporal effects of tactile enrichment on behavior and DG neuron activation. a** Experimental design for the effects of tactile enrichment on memory and anxiety. STE, short-term tactile enrichment; OLT, object location task; EPM, elevated plus maze. $n = 10$ mice per group. **b** At 1 week but not 4 weeks after tactile enrichment, improvements in spatial memory performance were still detectable. **c** At 1 week after tactile enrichment, anxiety level was reduced as shown by increased percentage of time in open arms and number of head dips in the elevated plus maze. **d** Experimental design for the temporal effects of tactile enrichment on DG neuron activation. $n = 7$ mice per group. **e** Representative plots of c-fos+ neurons superimposed on empty space distance maps of dDG and vDG. **f** The density of c-fos+ neurons in dDG was similar among groups, whereas the density of activated vDG neurons was increased in the TEE group compared to controls. **g** Activated neurons in dDG and vDG were more clustered after tactile enrichment ended, but became dispersed again 1 week later. *$P < 0.05$, **$P < 0.01$, ***$P < 0.001$. Data in bar charts are presented as mean ± SEM. Detailed statistics are provided in Supplementary Table 2. Source data are provided as a Source Data file.

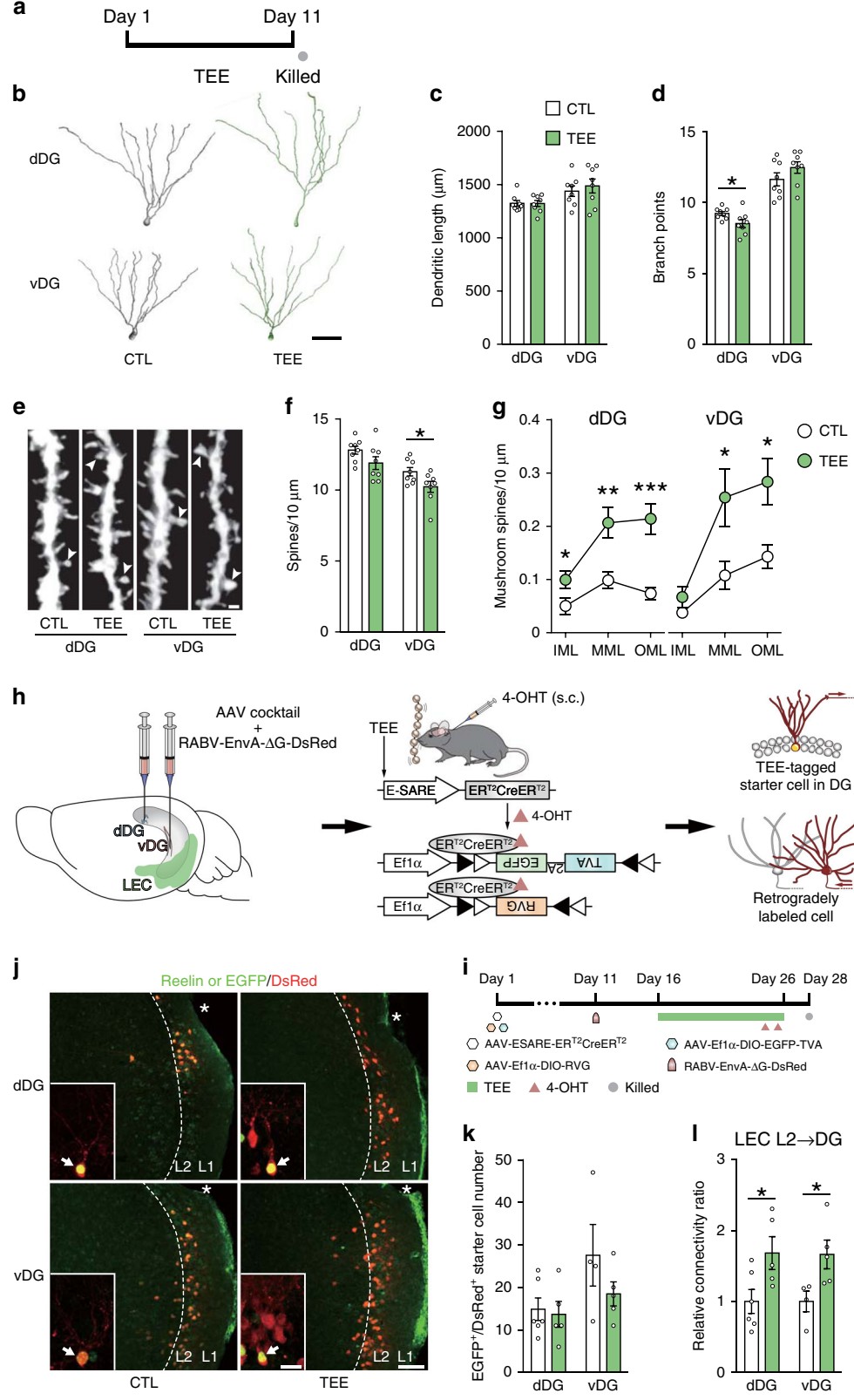

**S1-innervated LEC neurons modulate the behavioral effects of tactile enrichment**. To evaluate the influence of S1 input on the activation of LEC and DG neurons, the hM3Dq-expressing AAV was injected to the S1 of Scnn1a-Cre mice that expressed Cre in S1 layer 4 excitatory neurons[36] (Fig. 7a). Both acute and chronic

chemogenetic activation of S1 layer 4 neurons increased c-fos[+] neuron density in LEC layer 2 and vDG (Fig. 7b, c), and mimicked the effects of tactile enrichment on the spatial pattern of activated neurons in dDG and vDG (Fig. 7d, e). In addition, mice with acute S1 stimulation showed increased activation of

**Fig. 4 Effects of tactile enrichment on DG neuron structural plasticity and synaptic connectivity. a** The timeline of the morphological experiment $n = 8$ mice per group. **b** Representative reconstructions of dDG and vDG granule cells from the CTL and TEE groups. Scale bar = 50 μm. **c** Dendritic length of dDG and vDG neurons was comparable between groups. **d** Tactile enrichment subtly reduced the number of branch points in dDG neurons. **e** Representative images of dendritic segments in the outer molecular layer of dDG and vDG. Arrowheads indicate mushroom spines. Scale bar = 1 μm. **f** Tactile enrichment reduced total spine density in vDG neurons. **g** Quantification of mushroom spine density in the inner, medial and outer molecular layers (abbreviated as IML, MML, and OML) of the DG. Tactile enrichment increased the number of mushroom spines in all sublayers of dDG and in MML and OML of vDG. **h** Illustration of the strategy to retrogradely label lateral entorhinal cortex (LEC) neurons that project to TEE-tagged neurons in dDG or vDG. 4-OHT, 4-hydroxytamoxifen. s.c., subcutaneous. **i** The timeline of experiment. After viral delivery to dDG or vDG, mice received subcutaneous injections of 4-OHT on days 24 and 25 and were killed on day 28. **j** Representative images show reelin (a marker of entorhinal layer 2 fan/stellate cells) and DsRed (expressed by retrogradely labeled cells) expression in the LEC. Asterisks indicate the rhinal fissure. Insets show EGFP and DsRed double-labeled starter cells (indicated by arrows) in the DG. Scale bar is 100 μm for LEC and 20 μm for DG. **k** The number of starter cells in dDG and vDG was comparable between groups. **l** Relative connectivity ratio was calculated as the number of retrogradely labeled DsRed[+] cells in LEC layer 2 divided by the number of starter cells in DG. Tactile enrichment increased presynaptic input from LEC layer 2 neurons to TEE-tagged neurons in dDG and vDG. *$P < 0.05$, **$P < 0.01$, ***$P < 0.001$. Data are presented as mean ± SEM. Detailed statistics are provided in Supplementary Table 2. Source data are provided as a Source Data file.

LEC layer 3 neurons, and had more c-fos[+] neurons in LEC layers 5-6 and dDG compared to the chronic activation group.

To further examine the involvement of S1-innervated LEC neurons in tactile enrichment effects, the anterograde Cre-expressing AAV and the hM4Di-expressing AAV were injected into the S1 and LEC, respectively. Drinking water supplied with or without CNO was provided during the 10-day tactile enrichment. Before and at 1 week after tactile enrichment, mice were tested in object location and elevated plus maze tasks (Fig. 7f). In both behavioral tests, a significant enrichment × treatment interaction was found. At 1 week after tactile enrichment, mice with chemogenetic inhibition of S1-innervated LEC neurons exhibited impaired spatial memory performance and increased anxiety level compared to mice provided with normal drinking water (Fig. 7g–i). These data suggest that S1-innervated LEC neurons, which indirectly or directly target TEE-tagged DG neurons, are indispensable for the behavioral effects of tactile enrichment.

**Adulthood tactile enrichment attenuates early life stress-induced memory loss and anxiety-related behavior.** Exposure to unpredictable maternal care during critical developmental periods leads to negative cognitive and emotional outcomes[37]. We applied an early life stress paradigm[38], in which mouse pups received fragmented maternal sensory (including tactile) inputs during postnatal days 2–9, and investigated whether tactile enrichment in adulthood could ameliorate the adverse consequences of this early life stress (Fig. 8a). Mice with early stressful experience performed significantly worse than controls in object recognition tasks, whereas adult tactile enrichment restored the ability to discriminate object novelty, location, and temporal order in stressed mice (Fig. 8b–e). All groups of mice performed similarly in the open field test (Supplementary Fig. 12a) and spent comparable time in the light chamber of the light-dark box (Supplementary Fig. 12b). Nevertheless, stressed mice visited the anxiogenic light chamber less frequently, indicative of increased anxiety level, which could be normalized by adult tactile enrichment (Fig. 8f). In the elevated plus maze test, a significant main effect of tactile enrichment on the percentage of time in open arms was found (Fig. 8g). Moreover, stressed mice had more entries to the closed arms than controls, while the two enrichment groups showed similar performance (Supplementary Fig. 12c). These data demonstrate that adult tactile enrichment attenuates the negative effects of early life stress on memory and anxiety.

We further used Golgi staining to examine how early life stress would interact with adult tactile enrichment to shape the plasticity of dentate granule cells. In the dDG, we found a significant main effect of tactile enrichment on dendritic length

(Fig. 8h, i) and stubby spine density (Supplementary Fig. 12d, e). Moreover, early life stress reduced dendritic complexity, which could be normalized after tactile enrichment (Fig. 8j). No main or interaction effect on total, mushroom nor thin spine density was detected in dDG granule cells (Supplementary Fig. 12e, f). In the vDG, a main effect of tactile enrichment was observed on the density of total, mushroom and thin spines (Fig. 8k–m). Dendritic length and complexity of vDG granule cells were comparable among groups (Supplementary Fig. 12g–j). Therefore, the structural plasticity of dDG and vDG neurons is differentially modulated by the interplay between early life stress and adult tactile enrichment, which in turn contributes to the behavioral phenotypes.

## Discussion

Neural circuits in the adult brain undergo dynamic reorganization in response to sensory experience. Such experience-dependent plasticity underlies behavioral adaptation. We showed that repeated tactile enrichment positively influenced cognitive and emotional behavior, accompanied by activation pattern changes and dendritic spine remodeling in dDG and vDG. The beneficial behavioral effects of tactile enrichment were modulated by TEE-tagged DG neurons as well as LEC neurons that relay integrated information from S1 to DG. Moreover, adulthood tactile enrichment ameliorated early life stress-induced abnormalities. These findings reveal the neural mechanisms of enriched tactile experience-induced effects both under normal conditions and after stress exposure.

Environmental enrichment exerts beneficial effects on spatial memory and emotion regulation[39], and enhances synaptic plasticity in the primary sensory cortices[28]. We found that, compared to V1 and Au1, the thalamo-recipient layer 4 in S1 was remarkably activated by environmental enrichment, raising the possibility that the tactile component of multimodal inputs drives experience-dependent plasticity in S1-related circuits to modulate behavior. An increase in DG neuronal activation by multimodal enrichment implies the potential link of DG to such circuits. Indeed, somatosensory deprivation or stimulation decreases or increases neuronal activity in S1 respectively[24,40], while a similar bidirectional, tactile input-dependent activation of dDG neurons has been reported[23,24]. Similar to multimodal enrichment, unimodal tactile enrichment increased neuronal activation in the barrel field and trunk regions, which occupy ~70% of the S1[41], and in the vDG. In the dDG, tactile enrichment altered the spatial distribution pattern instead of the density of activated neurons. In addition, both acute and chronic chemogenetic activation of S1 layer 4 neurons increased activated neuron density in dDG and vDG, and mimicked the influence of tactile enrichment on the spatial pattern of activated neurons. These results indicate that

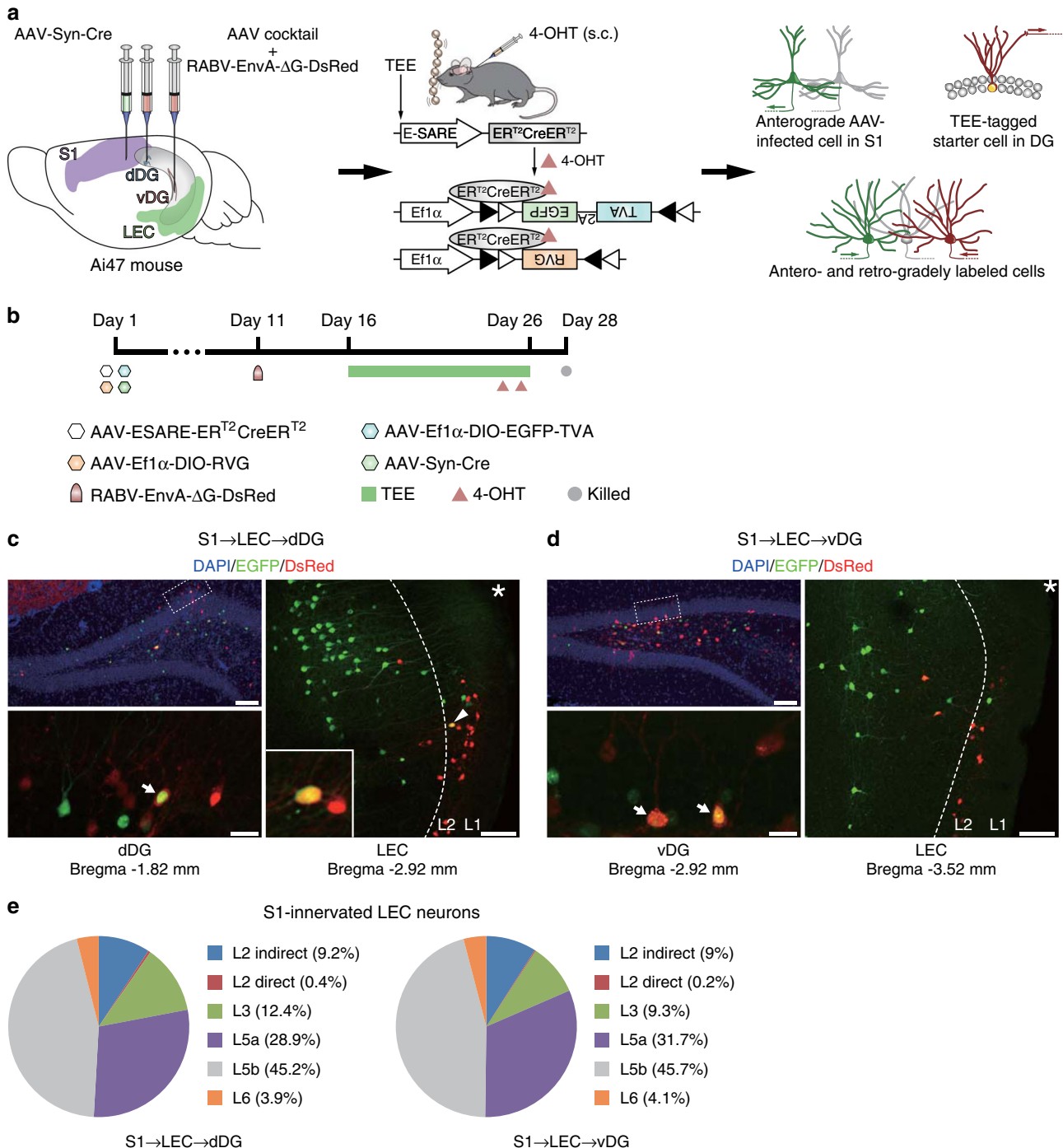

**Fig. 5 Transsynaptic tracing of S1-innervated LEC cells that project to TEE-tagged DG neurons. a** Illustration of the strategy for simultaneous anterograde and retrograde monosynaptic tracing. **b** The timeline of experiment. $n = 3$ and 2 mice for dDG and vDG experiments respectively. **c**, **d** Upper left panels show DG cells stained by DAPI, EGFP and DsRed. Boxed regions were imaged at a higher magnification and are presented in lower left panels, where arrows indicate EGFP and DsRed double-labeled starter cells. Right panels show anterogradely labeled S1-innervated cells (EGFP+) and retrogradely labeled (**c**) dDG- or (**d**) vDG-projecting cells (DsRed+) in the LEC. The arrowhead indicates an LEC layer 2 cell (magnified in the inset) that both received S1 input and projected to TEE-tagged dDG neurons. Scale bars are 100 μm for upper left and right panels and 20 μm for lower left panels. Asterisks indicate the rhinal fissure. **e** Pie charts depict the percentage of S1-innervated LEC layers 2–6 neurons that indirectly or directly targeted TEE-tagged dDG and vDG neurons. Source data are provided as a Source Data file.

enriched tactile experience modulates DG neuron activation, the extent of which depends on the nature, intensity and duration of tactile inputs.

Multimodal enrichment increases adult hippocampal neurogenesis, which contributes to the enrichment effects on memory and anxiety[28,34]. We found that unimodal tactile enrichment did not affect the proliferation and differentiation of adult-born cells in DG. The expression levels of stress-related molecules and markers for mature granule cells and inhibitory synapses remained unchanged in mice with tactile enrichment. Dendritic outgrowth and spine formation in newly generated DG neurons, hallmarks of morphological maturation[42], were also not affected

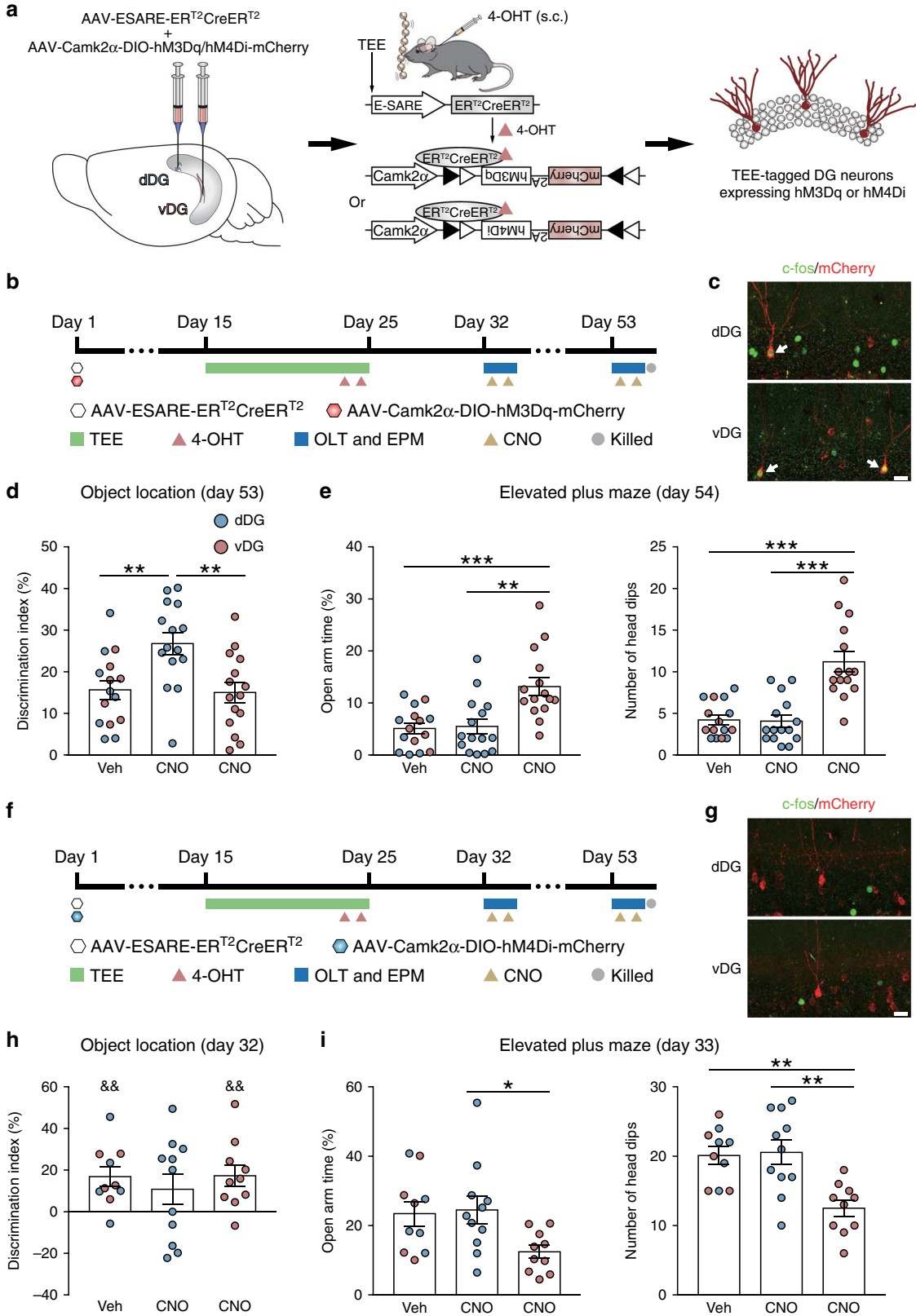

by tactile enrichment. These results suggest that the behavioral effects of tactile enrichment are not dependent on adult hippocampal neurogenesis. Since most tactile enrichment-activated DG neurons were calbindin+ granule cells, tactile enrichment mainly targets and remodels mature DG neurons to modulate memory and anxiety level.

We found that tactile enrichment induced temporally specific effects on DG neuronal activation, memory performance, and emotional responses. Moreover, recurrent tactile enrichment increased mushroom spine density in Golgi-stained dentate granule cells with mature morphology. It is plausible that sustained tactile experience-driven activity of dDG and vDG neurons

**Fig. 6 Effects of chemogenetic manipulation of TEE-tagged dDG or vDG neurons on memory and anxiety. a** Illustration of the strategy to express hM3Dq or hM4Di in TEE-tagged dDG or vDG neurons for chemogenetic manipulation. **b** The timeline of the chemogenetic activation experiment. CNO, clozapine-N-oxide. $n = 15$ mice per group. **c** Representative images from the dDG-CNO group (upper panel) and the vDG-CNO group (lower panel) show c-fos and mCherry immunostaining in dDG and vDG respectively. Arrows indicate activated granule cells that were labeled during tactile enrichment. Scale bar = 20 μm. **d, e** At 4 weeks after tactile enrichment, chemogenetic activation of TEE-tagged dDG but not vDG neurons enhanced performance in the object location task (**d**), whereas activation of TEE-tagged vDG but not dDG neurons increased the time in open arms and the number of head dips in the elevated plus maze (**e**). **f** The timeline of the chemogenetic inhibition experiment. $n = 10$–11 mice per group. **g** Representative images show the absence of colocalization between c-fos and mCherry in dDG or vDG after CNO administration. Scale bar = 20 μm. **h, i** At 1 week after tactile enrichment, mice with inactivation of TEE-tagged dDG but not vDG neurons failed to discriminate the objects, indicative of impaired spatial memory (**h**). Inactivation of TEE-tagged vDG but not dDG neurons reduced the time in open arms and the number of head dips in the elevated plus maze (**i**). *$P < 0.05$, **$P < 0.01$, ***$P < 0.001$. &&$P < 0.01$ for pairwise comparisons between the percentage of time exploring the relocated and the stationary objects. Data are presented as mean ± SEM. Detailed statistics are provided in Supplementary Table 2. Source data are provided as a Source Data file.

is needed to stabilize structural modifications and modulate behavior. In line with this reasoning, we found that tactile enrichment recruited subpopulations of DG neurons to support memory retrieval and reduce anxiety level. Although the functional organization of the hippocampus along the dorsoventral axis appears to be more graded than discrete[43], dDG and vDG have been shown to modulate memory and anxiety respectively[25,26]. Using functional labeling of neurons and chemogenetics, we manipulated the activity of previous tactile experience-activated neurons in dDG or vDG. At 4 weeks after tactile enrichment when its positive effects disappeared, reactivation of these neuronal ensembles in dDG or vDG could reproduce memory-enhancing or anxiolytic effects. In comparison, suppression of their activity did not impair behavioral performance. An opposite scenario was observed 1 week following tactile enrichment when its effects were still present: activation of TEE-tagged dDG or vDG neurons could not further affect memory nor anxiety, whereas inactivation of these neurons diminished the positive effects of tactile enrichment. On the one hand, the ceiling effect of DG activation as well as the floor effect of its inhibition suggest that TEE-tagged DG neurons play a modulatory role in cognitive and emotional behavior, and these neurons take effect only in a certain range. On the other hand, structural remodeling in these neuronal ensembles likely retunes DG output, which in turn coordinates the activity of CA3 and downstream structures to influence behavior. In addition, it should be mentioned that when increasing the sample size of the tactile enrichment group, dendritic branch point number in dDG neurons was reduced, while dendritic length and intersections at concentric circles remained comparable to control mice. Such discrepancy among dendritic parameters has been observed in previous studies[44,45]. However, considering that the change in dendritic branch points was subtle, tactile enrichment mainly induces dendritic spine remodeling in DG neurons.

The S1 does not directly innervate DG neurons[46], but is reciprocally connected with the LEC[19–21] that conveys both nonspatial and spatial information about the content of an experience to the hippocampus[47,48]. Our viral tracing results showed that most S1-innervated LEC neurons indirectly targeted TEE-tagged dDG and vDG neurons, possibly through the connection from LEC layer 5b to layer 2[49]. Such organization may allow tactile inputs to be further processed and integrated in the LEC before entering the DG. Following tactile enrichment, each node of the S1 → LEC → DG pathway underwent experience-dependent plasticity, as shown by enhanced activation of S1 neurons, increased input from LEC neurons and spine remodeling in DG granule cells. The interaction among S1, LEC and DG and its role in tactile enrichment-induced effects were further revealed by the results that stimulation of S1 layer 4 neurons increased LEC and DG neuronal activation, while silencing S1-innervated LEC neurons abolished the behavioral effects of tactile enrichment.

Considering their importance in memory[48,50], enhanced synaptic plasticity in LEC and dDG neurons driven by increased somatosensory inputs may explain cognitive improvement by tactile enrichment. In addition, vDG granule cells have been shown to exert anxiolytic effects[25], while a role of the LEC in innate and acquired defensive behaviors has been suggested[51,52]. Together with our data, it is possible that enhanced LEC-vDG connectivity and plasticity by tactile enrichment may modulate cognitive appraisal of threat to reduce anxiety[53].

Emerging evidence suggests that somatosensory processing is impaired in cognitive and stress-related disorders. For example, peripheral stimulation-evoked S1 activity is increased in patients with mild cognitive impairment[8], whereas innocuous touch-induced S1 response is decreased in patients with post-traumatic stress disorder[9]. In addition, exposure to neonatal or adolescent stress elevates basal/evoked activity and increases spine elimination in the mouse S1[10,54]. In complement to these findings, we showed that the adverse impact of early life stress on memory and anxiety-related behavior could be ameliorated by tactile enrichment in adulthood. Furthermore, early life stress and adult tactile enrichment differentially modulated the structural plasticity of dDG and vDG neurons. These findings suggest the potential of tactile enrichment to improve cognition and reduce anxiety in healthy and stressed individuals. Disrupted connectivity and/or activity of the pathway from the S1 to the dDG/vDG likely modulate or even exacerbate the effects of early life stress on cognition and emotional responses, which merits further investigation.

The current study has several other limitations. First, in vivo imaging, electrophysiological recording and molecular profiling of activated neurons in DG and other hippocampal subfields will help identify the dynamics and molecular mechanisms of neuronal plasticity evoked by tactile enrichment. Second, as LEC also receives inputs from other primary sensory cortices, future studies are needed to compare how LEC neurons process information from different sensory modalities to influence DG neuronal plasticity, memory and anxiety. Third, most experiments were performed in males. The interaction between sex and tactile experience on cognitive and emotional behavior remains to be examined.

In summary, our study reveals the critical role of dentate granule cells along the dorsoventral axis in modulating the beneficial effects of tactile enrichment, and identifies a neural pathway bridging somatosensory processing to memory and anxiety.

# Methods
**Animals**. C57BL/6N mice (SLAC Laboratories), Rosa26-GFP reporter mice (Allen Institute for Brain Science) and Scnn1a-Cre mice (stock number 009613, Jackson Laboratory) were used. Adult mice were group-housed (3-4 per cage) under a 12:12-h light/dark cycle (lights on at 8 a.m.) and controlled temperature (22 ± 2 °C) and humidity (50 ± 10%) conditions with free access to food and water, and were

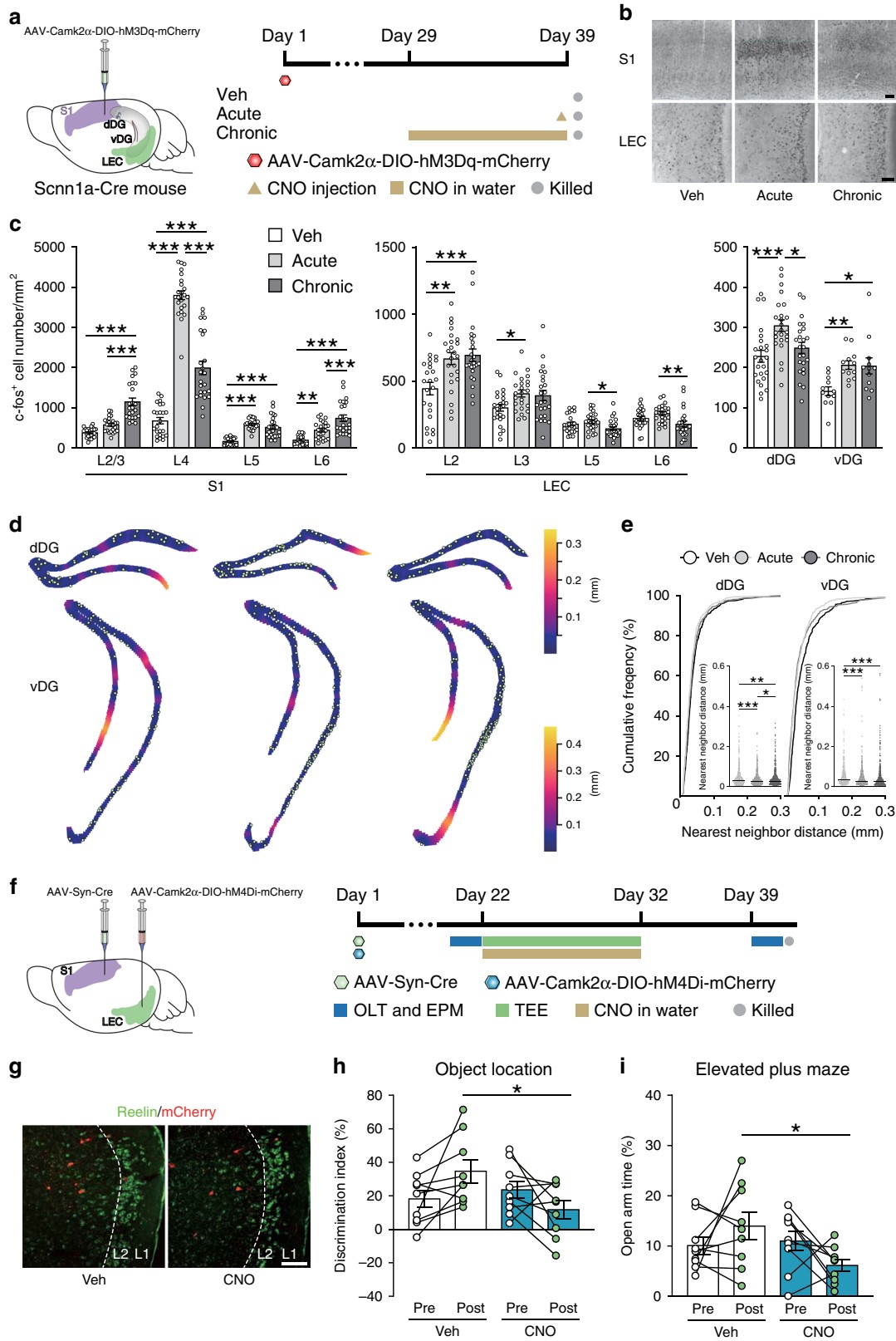

randomly assigned to each group. The experiments were approved by the Animal Advisory Committee at Zhejiang University and performed in compliance with the National Institute of Health's Guide for the Use and Care of Laboratory Animals.

**Environmental enrichment procedure**. The environmental enrichment procedure was modified from established protocols[55] to provide multimodal stimuli. Adult male mice (2 months old) were housed in groups of 6 in large cages (61 × 43.5 × 21 cm³) containing a nest box, a running wheel, plastic tubes, Lego bricks, 4.8 g of Nestlets (Indulab), and corncob beddings for 10 days. Control males were individually housed in standard-sized cages (28 × 17 × 14 cm³) with bedding material only.

**Tactile enrichment procedure**. The mouse model of unimodal tactile enrichment was inspired by a previous study[56]. The establishment of this model is detailed in

**Fig. 7 Effects of chemogenetic activation of S1 layer 4 excitatory neurons on LEC and DG neuron activation and the involvement of S1-innervated LEC neurons in tactile enrichment effects. a** Illustration of viral injection to the S1 of Scnn1a-Cre mice and the timeline of experiment. $n = 3$ mice per group. **b** Representative photomicrographs show c-fos$^+$ neurons in the S1 and LEC of mice with vehicle (Veh), acute CNO (Acute), or chronic CNO (Chronic) treatment. Scale bars = 100 μm. **c** Acute chemogenetic activation of excitatory neurons in S1 layer 4 increased c-fos$^+$ neuron density in S1 layers 4-6, LEC layers 2-3, dDG, and vDG, while chronic activation of S1 layer 4 increased c-fos$^+$ neuron density in S1 layers 2-6, LEC layer 2 and vDG. **d** Representative plots of c-fos$^+$ neurons superimposed on empty space distance maps of dDG and vDG. **e** Both acute and chronic activation of S1 layer 4 neurons altered neuronal activation pattern in dDG and vDG. **f** Illustration of the strategy to express hM4Di in S1-innervated LEC neurons and the timeline of experiment. $n = 9$ mice per group. **g** Representative images show reelin and mCherry expression in the LEC. Scale bar = 100 μm. **h, i** Chemogenetic silencing of S1-innervated LEC neurons abolished the positive effects of tactile enrichment on spatial memory performance (**h**) and anxiety-related behavior (**i**). *$P < 0.05$, **$P < 0.01$, ***$P < 0.001$. Data in bar charts are presented as mean ± SEM. Detailed statistics are provided in Supplementary Table 2. Source data are provided as a Source Data file.

Supplementary Fig. S2. Adult female mice (2 months old) were used for model establishment, whereas adult males (2–2.5 months old) were used for other experiments. At 1 week before tactile experience enrichment, mice were individually housed. Tactile enrichment was always started at 1 h before the dark phase and performed in standard-sized cages, in which a bead curtain consisting of a row of bead strings, 4.8 g of Nestlets and 500 ml of birch shavings were provided. Custom-made beads were 0.8 cm in diameter, made of transparent crystal glass, and smooth on surface. Each bead string consisted of 9 beads connected by a piece of insulated wire with a total weight of 11.51 ± 0.02 g. The mechanical force to move 1 bead string for 15–45° was 0.13–0.19 N, which could be imposed on the body surface of freely exploring mice. The bead curtain contained 19 bead strings and divided the enrichment cage into three zones: the nest zone (where the nesting material was provided; $7.5 \times 17 \times 14$ cm$^3$), the curtain zone ($1 \times 17 \times 14$ cm$^3$), and the feeding zone (where food and water were provided; $19.5 \times 17 \times 14$ cm$^3$). Such arrangement allowed mice to rest in the nest zone and consume food and water in the feeding zone, thus increasing curtain exploration.

For short-term tactile enrichment, mice were placed in the enrichment cages for 1.5 days. For prolonged tactile enrichment, mice stayed in the enrichment cages for 10 or 20 days, and cages were changed every 5 days. A cohort of mice was housed in tactile enrichment cages with varied rows of bead curtain (a half row, 1 row, or 3 adjacent rows) for 10 days to compare the density of c-fos$^+$ dentate granule cells. Control mice remained singly housed in standard cages with bedding material only. In several experiments, group-housed mice (4 per cage) were also included. After tactile enrichment, mice were either transferred to standard cages for further analyses or killed.

**Early life stress procedure**. The limited bedding and nesting (LBN) material paradigm, a naturalistic and robust early life stress model, was performed[38,57]. On the morning of postnatal day (P) 2, litters were culled to 6-8 pups with equal numbers of males and females whenever possible. Control dams were provided with a sufficient amount of nesting material (4.8 g of Nestlets) and 500 ml of birch shavings. LBN dams were provided with 1.2 g of Nestlets that was placed on an aluminum mesh platform (McNichols). The stress procedure ended on the morning of P9. Male mice were weaned on P28 and group-housed.

**Behavioral testing**. To compare the exploration pattern in the tactile enrichment and control cages, mice were individually placed into respective cages for 1.5 days. The experiment was started at 7 p.m. on day 1, and home-cage activity was monitored at 9:00-9:30 p.m. on day 2 and 6:30–7:00 a.m. on day 3. The distance traveled in each zone and the number of zone entries were scored by ANY-maze 5.14 (Stoelting). Memory and anxiety tests were performed between 9 a.m. and 4 p.m. and scored by ANY-maze[57,58].

**Object recognition tasks**. Object recognition tasks were performed in the open field arena ($50 \times 50 \times 50$ cm$^3$) under low illumination (10 lux). At 24 h prior to the acquisition phase, mice were habituated to the testing environment without objects or spatial cues (pre-training). During testing, objects were placed in the corners of the arena, 5 cm from the walls. Each pre-training, acquisition and retrieval trial lasted 10 min. The percentage of time exploring each object and the discrimination index (DI) in the retrieval trial were calculated. In addition, the object preference ratio in the acquisition trials was calculated and summarized in Supplementary Fig. 13.

For the novel object recognition task, both the pre-training and the testing procedures consisted of 2 trials separated by an intertrial interval (ITI) of 4 h. No spatial cue was provided during testing. In the acquisition phase, two identical triangular pyramids were presented. In the retrieval phase, one of the pyramids (the "known" object) was replaced by a culture flask filled with sand (the "novel" object). DI was calculated as 100% × (time with the novel object−time with the known object)/time with both objects.

For the object location task, both the pre-training and the testing procedures consisted of 2 trials separated by an ITI of 4 h. Prominent spatial cues were provided during testing. In the acquisition phase, two identical circular cones were

presented. In the retrieval phase, mice were presented with a non-displaced object and a relocated one. DI was calculated as 100% × (time with the relocated object−time with the stationary object)/time with both objects.

For the temporal order task, both the pre-training and the testing procedures consisted of 3 trials separated by 60-min ITIs. No spatial cue was provided during testing. In acquisition phases 1 and 2, two cubes and two cylinders were presented respectively. In the retrieval phase, one cube (the "remote" object) and one cylinder (the "recent" object) were presented. DI was calculated as: 100% × (time with the remote object−time with the recent object)/time with both objects.

For the object-in-place task, the pre-training included 1 trial. During testing, prominent spatial cues were provided and the ITI was 30 min. In the acquisition phase, mice were presented with four objects different in shape. In the retrieval phase, two of the objects exchanged positions. DI was calculated as 100% × (time with the relocated objects−time with the stationary objects)/time with all objects.

**Open field test**. Mice were placed in the open field arena made of gray polyvinyl chloride and evenly illuminated at 30 lux. The time and distance traveled in the center zone ($25 \times 25$ cm$^2$) and total distance traveled were recorded for 5 min.

**Light-dark box test**. Mice were placed in the dark chamber ($15 \times 20 \times 25$ cm$^3$, 10 lux), which was connected to the brightly illuminated chamber ($30 \times 20 \times 25$ cm$^3$, 650 lux) by a 4-cm long tunnel. During the 5-min test, the time spent in and the number of full entries to the light chamber were measured.

**Elevated plus maze test**. The elevated plus maze was made of gray polyvinyl chloride with two opposing open arms ($30 \times 5 \times 0.5$ cm$^3$, 40 lux) and two opposing enclosed arms ($30 \times 5 \times 15$ cm$^3$, 10 lux) connected by a central platform ($5 \times 5$ cm$^2$). Mice were placed in the center of the maze, facing an open arm, and allowed to explore for 5 min. The time spent in and the number of full entries to each arm as well as the number of head dips were recorded. The percentage of time in open arms was calculated as: 100% × time in open arms/time in all arms.

**Stereotaxic microinjection of viral vectors**. Detailed information for viral vectors used in this study is provided in Supplementary Table 1. At 1 week before surgery, mice were individually housed. On the day of stereotaxic surgery and viral microinjection, mice were anesthetized in the home cage by 1 ml of isoflurane and secured in a stereotaxic frame (RWD Life Science). The anesthesia was maintained using isoflurane-O$_2$ (1–1.5:100) inhalation. AAV (0.3 μl for each hemisphere) or RABV (0.1 μl for each hemisphere) solution was bilaterally delivered to the dDG (1.7 mm posterior to bregma, 0.8 mm lateral from midline, and 2.0 mm ventral from dura), vDG (3.4 mm posterior to bregma, 2.6 mm lateral from midline, and 2.3 mm ventral from dura), S1 (0.7 mm posterior to bregma, 3.0 mm lateral from midline, and 0.6 mm ventral from dura for virus validation and transsynaptic tracing experiments; 0.2 and 0.9 mm posterior to bregma, 2.8 mm lateral from midline, and 0.5 mm ventral from dura for chemogenetic experiments), or LEC (2.9 mm posterior to bregma, 3.9 mm lateral from midline, and 3.0 mm ventral from dura) via a glass micropipette connected to a Nanoject III microinjector (Drummond Scientific) over a 10-min period. To label adult-born dentate granule cells, engineered self-inactivating murine retrovirus (RV) expressing enhanced green fluorescent protein (EGFP) under the control of the Ubi promoter was produced and purified[59]. RV solution (1 μl per injection site) was bilaterally delivered to infect both dDG and vDG neurons (2.5 mm posterior to bregma, 2.5 mm lateral from midline, and 1.8 mm ventral from dura). After viral microinjection, the micropipette was left in site for 5–7 min.

**Whisker trimming and specificity validation of AAV-ESARE-ER$^{T2}$CreER$^{T2}$**. Adult male mice (2 months old) received bilateral microinjections of 1:1 mixed AAV-ESARE-ER$^{T2}$CreER$^{T2}$ and AAV-Camk2α-DIO-hM3Dq-mCherry (Vigene Biosciences) to the S1. At 4 weeks after virus injection, whiskers on one side of the face were manually clipped by micro scissors as close to the skin as possible. Mice then received a subcutaneous injection of 4-hydroxytamoxifen (4-OHT; 40 mg/kg; Macklin) dissolved in corn oil containing 20% ethanol. At 30 min after 4-OHT

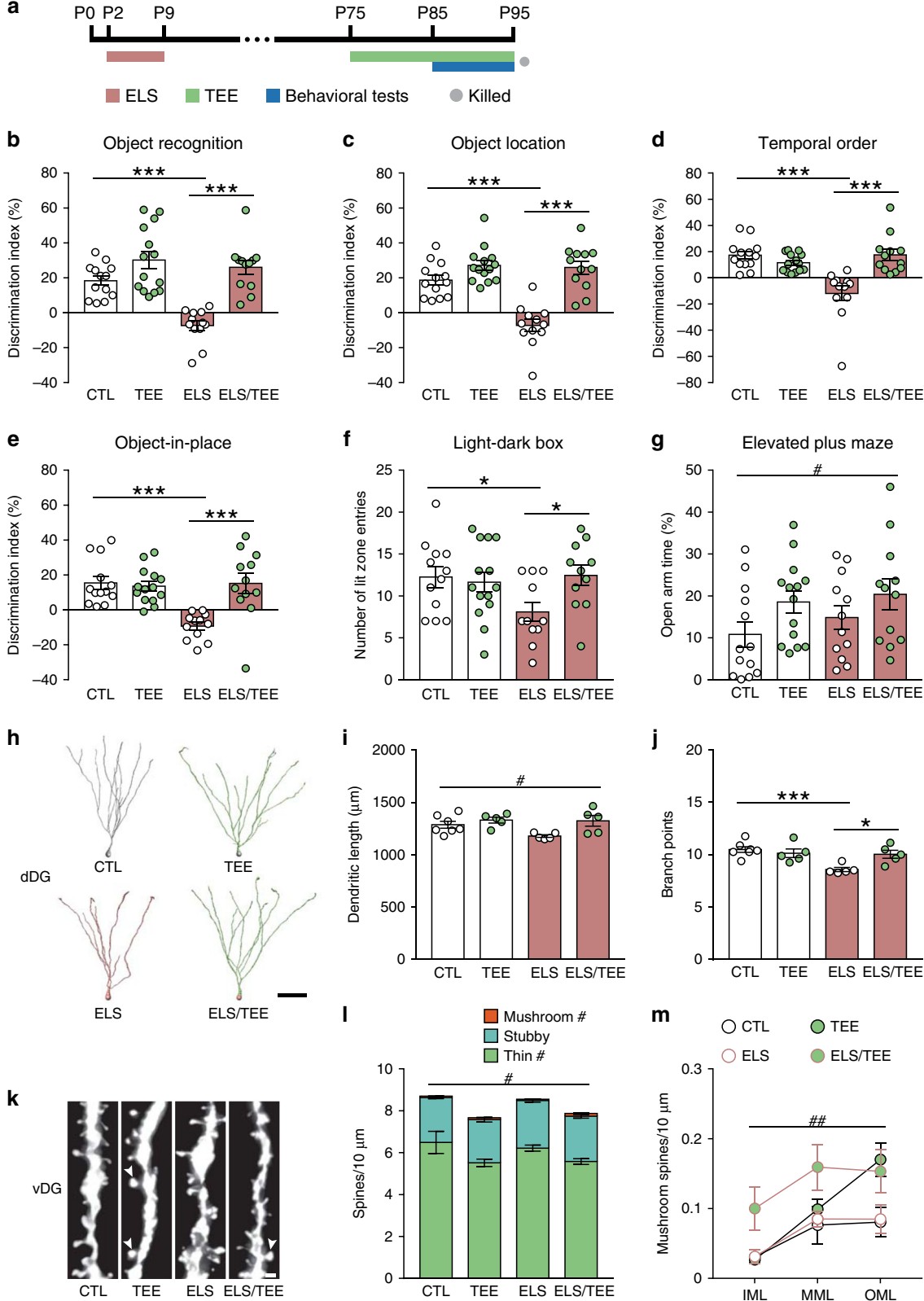

injection, whisker pads on both sides were gently stimulated by a foundation brush for 15 min. Mice were killed 1 week later.

**Retrograde and anterograde monosynaptic tracing.** To label the presynaptic inputs to TEE-tagged DG neurons, 1:1:1 mixed AAV-ESARE-ER^T2CreER^T2 and helper viruses (AAV-Ef1α-DIO-EGFP-TVA and AAV-Ef1α-DIO-RVG; BrainVTA Biotechnology) were bilaterally microinjected into the dDG or vDG of C57BL/6N

mice. Considering that the effects of tactile enrichment gradually decreased after enrichment ended and the invasiveness of stereotaxic viral injection, RABV-EnvA-ΔG-DsRed was delivered to the DG before, instead of during or after, the tactile enrichment procedure so that mice could recover from surgery and viral injection. It should also be noted that RABV can minimize tissue inflammation, escape the host immune response and evade antiviral immune clearance from the brain[60–62]. The lack of glycoprotein, a molecular determinant of pathogenicity, may also enable this modified RABV to escape immune recognition. Therefore, the injected

**Fig. 8 Effects of adulthood tactile enrichment on early life stress-induced behavioral and structural abnormalities. a** The timeline of experiment. ELS, early life stress. P, postnatal day. $n = 12–14$ mice per group for behavioral tests and 5–7 mice per group for structural analysis. **b–e** A significant stress × enrichment interaction on memory performance in novel object recognition (**b**), object location (**c**), temporal order (**d**), and object-in-place (**e**) tasks was found. Early life stressed mice failed to discriminate object novelty, location, and temporal order, whereas tactile enrichment in adulthood restored cognitive performance. **f** Adulthood tactile enrichment normalized the number of entries to the anxiogenic lit zone in stressed mice. **g** Tactile enrichment increased the time in open arms of the elevated plus maze. **h** Representative reconstructions of dDG granule cells. Scale bar = 50 μm. **i** Tactile enrichment increased dendritic length in dDG neurons. **j** Tactile enrichment abolished early life stress-induced reduction of dendritic branch point number in dDG neurons. **k** Representative images of dendritic segments in the outer molecular layer of the vDG. Arrowheads indicate mushroom spines. Scale bar = 1 μm. **l** Tactile enrichment reduced the density of thin and total spines but increased mushroom spine density in vDG neurons. **m** Tactile enrichment increased mushroom spine density in all sublayers of the vDG. $*P < 0.05$, $***P < 0.001$. $\#P < 0.05$, $\#\#P < 0.01$, enrichment effect. Data are presented as mean ± SEM. Detailed statistics are provided in Supplementary Table 2. Source data are provided as a Source Data file.

RABV-EnvA-ΔG-DsRed could remain in the injection sites for days. At 5 days after RABV injection, mice were either exposed to tactile enrichment or remained in standard cages for 10 days. Based on previous findings that the activation of E-SARE stabilizes after repeated exposure to the same enriched environment[63], the last 2 days of tactile enrichment were selected for neuronal labeling, during which mice received a daily subcutaneous injection of 4-OHT (40 mg/kg) at 1 h before the dark phase. Mice were killed 2 days after tactile enrichment ended (i.e., 4 days after the first 4-OHT injection). This survival time between the first 4-OHT injection and killing was determined according to previous reports[64–66].

For simultaneous retrograde tracing of TEE-tagged DG neuron-projecting cells and anterograde tracing of S1-targeted cells, AAV-ESARE-ER[T2]CreER[T2] and helper viruses were injected to the dDG or vDG of Ai47 mice, while the anterograde transsynaptic AAV-Syn-Cre (BrainVTA) was injected to the S1. The following procedures were the same as above.

**Functional labeling and chemogenetic manipulation of neurons**. To increase or suppress the activity of TEE-tagged DG excitatory neurons, AAV-ESARE-ER[T2]CreER[T2] was 1:1 mixed with either AAV-Camk2α-DIO-hM3Dq-mCherry or AAV-Camk2α-DIO-hM4Di-mCherry (Vigene Biosciences) and bilaterally injected to the dDG or vDG of C57BL/6 N mice. After 2 weeks of recovery, mice were exposed to tactile enrichment for 10 days, and received a daily subcutaneous injection of 4-OHT (40 mg/kg) during the last 2 days of enrichment. Another cohort of mice received daily 4-OHT injections during the last 5 days of enrichment to validate the results of the activation study. In addition, to examine the behavioral effects of chemogenetic activation of control dDG or vDG neurons, a cohort of mice that remained singly housed under standard conditions throughout the experiment received 2 daily 4-OHT injections and was tested. At both 1 week and 4 weeks after tactile enrichment ended, mice were examined in the object location and elevated plus maze tasks on 2 consecutive days. At 10 min before the retrieval phase of the object location task or the elevated plus maze test, mice received an intraperitoneal injection of either clozapine-N-oxide (CNO; 0.5 mg/kg, dissolved in 0.9% saline; MedChem Express) or vehicle (0.9% saline). Mice were killed at 90 min after the last test ended.

To increase the activity of excitatory neurons in S1 layer 4, AAV-Camk2α-DIO-hM3Dq-mCherry was bilaterally injected to the S1 (2 injection sites per hemisphere) of Scnn1a-Cre mice. Control mice and mice in the acute CNO treatment group were provided with normal drinking water and received an intraperitoneal injection of saline and CNO (0.5 mg/kg) respectively at 38 days after viral delivery. Mice in the chronic CNO treatment group were provided with CNO-containing water (0.25 mg/ml, replenished every 3–4 days) at 4 weeks after viral delivery for 10 days, and received an intraperitoneal injection of saline at the end of CNO treatment. The choice of CNO dosage was based on previous studies[33,58]. Mice were killed at 90 min after the intraperitoneal injection.

To suppress the activity of S1-innervated LEC neurons, AAV-Syn-Cre and AAV-Camk2α-DIO-hM4Di-mCherry were bilaterally injected to the S1 (2 injection sites per hemisphere) and LEC of C57BL/6N mice, respectively. At 3 weeks after viral delivery, mice were transferred to tactile enrichment cages and provided with either normal or CNO-containing drinking water (0.25 mg/ml) for 10 days. Before tactile enrichment stated and at 1 week after tactile enrichment ended, mice were examined in the object location and elevated plus maze tasks on 2 consecutive days. Mice were killed at 90 min after the last test ended.

To label neurons that were active under home-cage conditions or at different stages of tactile enrichment, 1:1 mixed AAV-ESARE-ER[T2]CreER[T2] and AAV-Camk2α-DIO-hM3Dq-mCherry were bilaterally delivered to the dDG. After 3 weeks of recovery, mice in enrichment groups received a subcutaneous injection of 4-OHT (40 mg/kg) on the first, fifth or tenth day of tactile enrichment, and were killed after 6 days of enrichment, immediately after enrichment ended, or at 1 week after enrichment ended respectively. Control mice received the 4-OHT injection under basal conditions and were killed 6 days later.

**Immunostaining**. Mice were anesthetized with sodium pentobarbital (200 mg/kg) and perfused with 0.9% saline followed by 4% buffered paraformaldehyde. Serial coronal sections (30 or 40 μm thick) were prepared through the S1 (Bregma 0.50 to

−1.58 mm), dDG (Bregma −1.58 to −3.08 mm) and vDG (Bregma −3.08 to −3.80 mm) using a cryostat (Leica). The following primary antibodies were used: rabbit anti-c-fos (1:10000, 226003, Synaptic Systems), guinea pig anti-c-fos (1:1000, 226004, Synaptic Systems), rabbit anti-minichromosome maintenance complex component 2 (MCM2; 1:1000, 4007, Cell Signaling), goat anti-doublecortin (1:1000, sc-8066, Santa Cruz), rabbit anti-calbindin (1:2000, CB-38a, SWANT), mouse anti-calbindin (1:2000, 300, SWANT), rabbit anti-vesicular GABA transporter (VGAT; 1:2000, 131013, Synaptic Systems), rabbit anti-mineralocorticoid receptor (MR; 1:500, ab64457, Abcam), rabbit anti-glucocorticoid receptor (GR; 1:1000, sc-1004, Santa Cruz), rabbit anti-early growth response 1 (Egr1; 1:1000, MA5-15008, Invitrogen), rabbit anti-EGFP (1:2000, sc-8334, Santa Cruz), mouse anti-reelin (1:5000, MAB5364, Millipore), and mouse anti-mCherry (1:5000, E022110-01, EarthOx).

Free-floating sections were labeled with primary antibodies at 4 °C (overnight). For immunohistochemistry, after primary antibody incubation and rinsing, sections were labeled with a peroxidase-conjugated goat anti-rabbit (undiluted, ZB-2301, Zhongshan Golden Bridge) or anti-mouse (undiluted, ZB-2305, Zhongshan Golden Bridge) secondary antibody at room temperature (2 h). The 3,3′-Diaminobenzidine (DAB) Horseradish Peroxidase Color Development Kit (Zhongshan Golden Bridge) was used for staining. For double-labeling of c-fos and mCherry, sections were sequentially incubated with the rabbit anti-c-fos and the goat anti-rabbit antibodies. Sigma FAST DAB with metal enhancer (D0426, Sigma-Aldrich) was used to yield a dark blue stain. Sections were then incubated with the mouse anti-mCherry antibody and the corresponding secondary antibody. The DAB Horseradish Peroxidase Color Development Kit was used to yield a brown stain. For immunofluorescence, after incubation with primary antibodies and rinsing, sections were labeled with Alexa Fluor 488- and/or 594-conjugated secondary antibodies at room temperature (3 h). The following secondary antibodies were used: donkey anti-rabbit Alexa Fluor 488 (1:2000, A-21206, Invitrogen), donkey anti-guinea pig Alexa Fluor 488 (1:2000, 706-545-148, Jackson ImmunoResearch), donkey anti-goat Alexa Fluor 488 (1:2000, A-11055, Invitrogen), donkey anti-mouse Alexa Fluor 488 (1:2000, A-21202, Invitrogen), donkey anti-rabbit Alexa Fluor 594 (1:2000, A-21207, Invitrogen), and donkey anti-mouse Alexa Fluor 594 (1:2000, A-21203, Invitrogen). Sections were then rinsed, mounted on glass slides (Citotest) and coverslipped with Vectashield (Vector Laboratories) containing 4′,6-diamidino-2-phenylindole (DAPI).

**Image acquisition and analysis**. Immunofluorescence images (1024 × 1024 or 1600 × 1600 pixel²) were obtained with an Olympus IX83–FV3000 laser scanning confocal microscope, and a ×10 (NA 0.40), a ×20 (NA 0.75), a 40× (NA 0.95), and a ×60 oil-immersion (NA 1.42) objectives were used. For spatial pattern analysis of c-fos⁺ neurons in DG, density analysis of c-fos⁺ neurons in S1 subregions and transsynaptic tracing experiments, images were obtained using an Olympus VS120 virtual slide scanning system. For immunoreactivity analysis and other cell counting experiments, images (4140 × 3096 pixel²) of at least three sections per mouse were acquired at ×100 with an Olympus BX61 microscope and the cellSens Dimension 2.2 software.

Image data were analyzed with the ImageJ 1.52e software (National Institute of Health). For the quantification of c-fos⁺ or Egr1⁺ activated cells, only mice without behavioral testing were included. For most analyses, results were first averaged by sections from an animal and then averaged by animals in a group. For c-fos⁺ neuron analysis in Scnn1a-Cre mice, due to the limited sample size (3 mice per group), images from all mice in each group (8 images for S1, LEC and dDG, and 4 images for vDG per mouse) were pooled and averaged. Cell counts were presented as the number of immunoreactive cells per mm³ (for MCM2 and doublecortin) or mm² (for others). Relative protein levels of calbindin, VGAT, MR, and GR were determined by the differences in optical density values between the region of interest (ROI) and corpus callosum (background) from the same section. The LEC-DG connectivity ratio, which reflected the amount of presynaptic input from LEC to TEE-tagged DG neurons, was calculated as the number of retrogradely labeled DsRed⁺ cells in LEC layer 2 divided by the number of EGFP and DsRed double-labeled starter cells in DG[67,68]. Results were normalized by taking the mean value of the control group as 100% or 1. For the proportion of doublecortin⁺ or

calbindin[+] neurons in activated DG cells and the reactivation of conditionally labeled dDG neurons, z-stack images at 2-μm intervals from 3–4 sections per region per mouse were analyzed. The reactivation rate was calculated as the percentage of c-fos[+] neurons in all mCherry[+] neurons[69].

For the density analysis of S1 c-fos[+] neurons in Fig. 1d, Fig. 7c and Supplementary Fig. 2g, the barrel field as well as adjacent dysgranular and forelimb/shoulder/neck/trunk regions were selected, and 7–8 serial coronal sections (Bregma −0.6 to −1.8 mm) from each mouse were analyzed manually. For detailed quantification of c-fos[+] neurons in layers 2-6 of the hindlimb, forelimb, barrel field, shoulder, and trunk regions of S1 in Supplementary Fig. 4, 8–12 serial coronal sections (Bregma −0.2 to −1.8 mm) from each mouse were included. The image was imported to ImageJ and segmented by the U-Net Segmentation plugin[70] (https://sites.imagej.net/Falk/) with a model fine-tuned from a pertained cell segmentation model (https://lmb.informatik.uni-freiburg.de/lmbsoft/unet/). The fine-tuned model, examples and instructions can be accessed on GitHub (https://github.com/unetzjuser/Finetuned-unet-model-for-neuron-detection). Segmented c-fos[+] neurons in S1 subregions were inspected, corrected and analyzed by a researcher blind to group allocation.

**Spatial pattern analysis of c-fos[+] neurons in the DG**. The spatial distribution pattern and density of c-fos-immunostained DG neurons from 4 dDG sections and 2-3 vDG sections per mouse were analyzed by ImageJ and R. Briefly, after importing the image to ImageJ, the dentate granule cell layer was delineated using the Polygon Selection tool, and ROI coordinates were obtained using Macro Recorder. The image was segmented using the U-Net Segmentation plugin with our fine-tuned model. After inspection and correction, the segmented image was binarized and the coordinates of c-fos[+] neurons were determined by the Analyze Particles tool. The coordinates of ROI and c-fos[+] neurons were then organized in Excel and transferred to R. The nearest neighbor distance, a measure of spacing between points that reflects spatial point pattern and interpoint interaction[71], and the density of DG c-fos[+] neurons were analyzed by the spatstat package of R[32]. The R codes for spatial pattern analysis can be accessed on GitHub (https://github.com/unetzjuser/Spatial-pattern-analysis-of-neurons).

**Golgi staining, imaging and the analysis of dendrites and spines**. Golgi-Cox staining was performed using the FD Rapid GolgiStain[TM] Kit (FD Neuro-Technologies). Briefly, brains were immersed in the Golgi-Cox solution for 14 days and transferred to the 30% sucrose solution for 2-5 days in the dark at room temperature. Serial coronal sections (120 μm thick) were prepared using a Leica VT1000S vibratome, mounted on Superfrost plus slides (Thermo Scientific), stained, and coverslipped. Z-stack images ($4140 \times 3096$ pixel[2] for dendrites and $2070 \times 1548$ pixel[2] for spines) were obtained using the Olympus BX61 microscope equipped with an automated ProScan stage (Prior Scientific). For dendrite analysis, fully impregnated dentate granule cells in the suprapyramidal blade (8-9 dDG neurons and 5-6 vDG neurons) were scanned using a 40× objective with a z-step size of 1 μm. For spine analysis, intact dendritic branches (1-3 branches per neuron; 8-9 neurons for dDG and 6-8 neurons for vDG) parallel to the section plane and ended close to the hippocampal fissure were scanned using a 60× oil-immersion objective with a z-step size of 0.5 μm.

For morphological analysis of RV-labeled newly generated dentate granule cells, coronal sections (120 μm thick) were prepared using the Leica VT1000S vibratome and processed. Images with a voxel size of $0.311 \times 0.311 \times 2$ μm[3] for dendrites and $0.132 \times 0.132 \times 0.5$ μm[3] for spines were obtained with the Olympus IX83–FV3000 confocal microscope and the ×40 and ×60 objectives, respectively. Dendritic spine images were deconvolved using the AutoQuant X3.0.4 software (Media Cybernetics). Since the number of fully labeled and morphologically intact dDG and vDG neurons was limited in each mouse, neurons in suprapyramidal and infrapyramidal blades from all mice in each group were pooled.

Dentate granule cells were reconstructed by the NeuronStudio software[72]. Total dendritic length, the number of dendritic branch points, and the number of intersections at concentric circles (20 μm apart) were measured. Dendritic spines were categorized as mushroom (protrusions with a small neck and a large head), stubby (protrusions closely connect to the dendritic shaft and lack a clear neck), and thin (long protrusions with a bulbous head) subtypes. Spines in the inner, medial and outer molecular layers of the DG were manually counted using NeuronStudio by a researcher blind to group allocation. Data were expressed as the number of spines per 10 μm of dendrite.

**Statistical analysis**. SPSS 22.0 (SPSS), GraphPad Prism 8.0 (GraphPad Software) and R 3.6.3 (http://www.r-project.org/) were used to perform statistical analyses. The normality of data distribution was assessed by the Shapiro–Wilk test and the homogeneity of variances was examined by Levene's test. For between-group comparisons, two-tailed Student's $t$ test and Mann-Whitney $U$ test were applied for normally and non-normally distributed data respectively. For multiple group comparisons, data were analyzed by analysis of variance (ANOVA) followed by Tukey's or Tamhane's post hoc test when appropriate. Differences in nearest neighbor distances among groups were evaluated using the Kruskal–Wallis test followed by Dunn's post hoc test when appropriate. In chemogenetic experiments, because no difference in behavioral performance was found between the dDG-

vehicle and the vDG-vehicle groups, data from these two groups were pooled. Correlations were assessed by Pearson correlation coefficient. Detailed statistical methods, statistical results, and sample size are summarized in Supplementary Table 2. Data are reported as mean ± SEM. Statistical significance was defined at $P < 0.05$.

**Reporting summary**. Further information on research design is available in the Nature Research Reporting Summary linked to this article.

**Data availability**
The data that support the findings of this study are available from the corresponding author upon reasonable request. The fine-tuned model for c-fos[+] cell segmentation is deposited on GitHub (https://github.com/unetzjuser/Finetuned-unet-model-for-neuron-detection). Source data are provided with this paper.

**Code availability**
The R codes for spatial pattern analysis of c-fos[+] neurons and examples are deposited on GitHub (https://github.com/unetzjuser/Spatial-pattern-analysis-of-neurons).

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

## Acknowledgements
The authors thank Dr. Xiang Yu and Dr. Xiaowei Chen for providing Scnn1a-Cre mice, and Dr. Haruhiko Bito for providing the E-SARE plasmid. We thank Ms. Xue Xu, Ms. Jia-Qi Guo, Ms. Meng Sun, Ms. Xiao-Yi Long, and the Core Facilities of Zhejiang University Institute of Neuroscience for technical assistance. This work was supported by the National Key Research and Development Program of China (2016YFA0501000) and the National Natural Science Foundation of China (81761138044 and 81971260).

## Author contributions
X.-D.W. conceived the project and designed the experiments. C.Wang, H.L., K.L., Z.-Z. W., C.Wu, J.-Y.Y., Q.G., P.F., and X.-X.W. performed the experiments. C.Wang, H.L., K.L., Z.-Z.W., and X.-D.W. analyzed the data. Y.-J.L. and X.-D.W. supervised the experiments. S.-M.D., H.W., Y.G., J.H., B.-X.P., M.V.S., and X.-D.W. wrote the paper.

## Competing interests
The authors declare no competing interests.

**Additional information**

