## [Peer Review File · Nature Communications]

Reviewers' comments:

Reviewer #1 (Remarks to the Author):

The manuscript from Wang and colleagues investigates the role of tactile modulation on memory and anxiety-like behaviour. It is a thorough, extensive, and multi-disciplinary study with many novel findings. The authors suggest that enriched tactile experience retunes the S1→LEC→dDG/vDG pathway and enhances DG neuron plasticity to modulate cognition and emotion.

This is a body of work potentially worthy of publication in such a high profile journal, but there are some shortcomings. Limitations in methodological design limit the conclusions substantially. Additionally, further experiments are needed in order to fully to reveal the neural mechanisms of tactile enrichment on cognition.

Specific comments below;

Major Comments:

1. The authors suggest that the data presented in figure 1c-f indicate that tactile input constitutes a major sensory source under MME, and raise the possibility of S1-associated neural circuits mediating the effects of MME on cognition and emotion. However, the data in Fig 1c-f is from multi-modal enrichment only, the authors should include in this figure (fig 1c-e) a tactile enrichment group so as to directly compare the effect of tactile experience from multi-modal experience on neuronal activation. (ie combine with supplementary figure 3e).

Furthermore, the authors should comment on the difference in magnitude between cfos activation in S1 following MME (fig 1c) and TEE (supp fig 3e). The cfos activation in L4 appears less in the TEE group than the control of the MEE?

2. It would be ideal to include an additional experimental group of artificial activation (chemogenetic or optogenetic) of S1 L4 on downstream activation of DG and LEC and compare to tactile induced activation.

3. The authors suggest that tactile inputs converge on the DG via LEC to shape cognitive process. Indeed, the authors showed that touch-tagged neurons in the DG received input from layer 2 LEC (Fig. 4j and Supplementary Fig. 7a). Thus, the manuscript would benefit from experimentally exploring whether inactivation of the LEC layer 2 during tactile enrichment blocks the downstream effects on DG activation and behaviour.

4. The authors persuasively show that their activity-dependent labeling method is indeed activity dependent. But they do not show that they have specificity for tactile stimulation. Would their interventions have the same behavioral effect if the stimulated home cage cells (labelled without tactile stimulation) or novel context engram cells? These are absolutely crucial control experiments.

5. How do the tactile enrichment effects on S1-LEC-DG plasticity compare to multimodal enrichment experience. The authors should include a MME group to the morphological analysis presented in figure 4 and 5.

Minor comments:

1. What did the pattern of neuronal activation look like in the vDG (figure 2i and j)? Include a graph of vDG pattern of neuronal activation or discuss relative to dDG.
2. The authors should include in supplementary figures the exploration times for the acquisition and test period for all the object recognition tasks.
3. The authors should offer some insight as to why enrichment may affect both the dorsal and

ventral DG SPB but only selectively affect the dorsal DG IPB (Fig 1e).

4. Line 204: Authors should rephrase sentence. They are not restoring memories. They have restored ability to discriminate. This is an important distinction.

5. Results section doesn't include any statistical information. It is in a table in the supplementary, however this is difficult to follow, can this be include in the text of results as is usually the case?

Reviewer #2 (Remarks to the Author):

In the present manuscript Wang C. et al. addressed the important question of how tactile enrichment modulates memory and anxiety in mice. Towards this, the authors developed a novel model of tactile enrichment (TEE) and combined immunohistological, morphological and behavioral analyses with state-of-the art approaches such as rabies-based connectivity tracing and chemogenetic manipulation of neuronal activity. They showed that tactile enrichment reduced anxiety and improved memory. Interestingly, TEE-induced activation of cortical somatosensory (S1) and dentate gyrus (DG) granule neurons promoted synaptic plasticity in DG neurons and increased lateral entorhinal cortex (LEC) connections to experience-activated DG neurons. Moreover, the authors used an elegant approach to specifically modulate experience-activated neurons in the ventral and dorsal DG. These well-conducted experiments confirmed the role of TEE-activated granule neurons in the modulation of memory and anxiety. Interestingly, the authors also showed that TEE is able to rescue memory deficits and ameliorates anxious behavior produced by early-life stress. This is an interesting and well-written paper which addresses important issues, but some concerns need to be addressed. Moreover, one of the described experiments seems to be impossible to have worked as the authors suggest.

1. One of the main issues of the manuscript concerns the characterization of the novel tactile enrichment model (TEE). Do the c-fos numbers correlate with the number of curtain zone entries? Does the half-curtain also activate c-fos? Moreover, can neuronal activation (c-fos+) be maximized by increasing the number of curtain rows (for example with 3 adjacent curtains)?

For curiosity, how specifically is this tactile circuit activating different sets of neurons, i.e are the same neurons activated every time the mice are stimulated? Is there a somatotopic representation of activated neurons in S1?

2. A serious technical issue concerns the rabies-tracing experiments. What is the rationale for injecting the rabies 2 weeks before 4-OHT? In principle, TVA expression (here, dependent on 4-OHT-induced activation of Cre) is required to render the cells susceptible to infection by the rabies. Thus, this referee thinks that the experiment as described cannot work as the rabies virus should remain in place for 2 weeks before being able to infect cells.

Also, efficient tracing requires (after 4-OHT injection) both recombination and spread of the rabies to occur; From the referee's experience 2 days would not be enough for that.

3. Also, there are issues with the experimental approach used to trace S1-to-LEC connections. If the AAV injected in S1 is indeed anterogradely transported, labelled axon terminals (rather than green somas) would be expected in LEC.

4. The quality of the images in Fig S1, S3E and G, S4, S6C and S8D need improvements. For all supplementary figures, higher resolution pictures would be required. The images shown in Fig 6C and G are insets of Fig S7F and G, thus should not be processed differently.

5. The authors should indicate precisely what is depicted in the correlation analysis (Fig 1F).

6. Was the morphometric analysis performed on activated neurons (E-SARE labelled DG-neurons)? This would be useful to specify.

7. From Fig S1 to conclude, the increase in c-fos positive cell number in S1 seem to be generalized to all layer (and not restricted to L4). Similarly for in V1 in Fig S1C, the increase seem to affect all layers.

Also, in Fig S3E, it is not possible to appreciate the increase in neuronal activation in L2-4 in S1.

8. The meaning of the pattern of activation analysis (Fig 2I and J) should be better explained.

Also, in the clustering analysis, the distance-path should follow the dentate gyrus shape. Instead

Fig 2J depicts a straight line crossing the hilus.

9. A proper analysis of adult-born neurons maturation (Fig S3H) would require thymidine-analogue experiments.

10. Fig S4E y-axis should indicate that it represents the "Percentage of calbindin positive neurons among activated cells".

11. In Fig 5E, how did the authors discriminate between direct and indirect connections? Do yellow cells correspond to direct connections? How did the authors label the indirect ones?

12. How many sections and cells were analyzed for IHC quantifications?

13. Could the authors explain why they used only males in their study, excepted for the TEE model establishment where they used only females?

Signed:

Benedikt Berninger, King's College London, UK

Reviewer #3 (Remarks to the Author):

Wang et al. propose an intriguing role of prolonged tactile stimulation with cognitive enhancements and anxiolytic behavior. The authors demonstrate that 1) tactile stimulation causes structural and activity modifications in the dentate gyrus (DG), accompanied by increased lateral entorhinal cortex connections to touch-associated DG neurons 2) manipulating activity of the touch-associated neurons bi-directionally modifies the behavior and anxiety effects, and 3) tactile stimulation reverses some of memory deficits and anxiety-like phenotypes associated with early life stress. The experiments are well designed, and the major strengths of this paper are its novelty and combination of powerfully selective technical approaches. Unfortunately, however, the study suffers from the lack of critical controls. Ultimately, the key conclusions that tactile enrichment strengthens an S1->LEC->DG circuit mediating recruitment of DG cells, and that DG cells activated by enrichment selectively control behavior, requires overinterpretation of the findings. My major and minor comments are detailed below.

Major comments:

- It is difficult to conclusively say that the increased c-Fos related to the multi- or unimodal tactile stimulation paradigms is associated with the touch sensation per se, or to a variety of other behaviors that could be related to interacting with the objects (e.g. increases in locomotion or exercise, positive affective associations to the objects, experiencing unique "contexts" generated by segmenting the cage into separate zones). While this does not necessarily undermine the results of the paper, it does muddy the water on identifying the kinds of sensory experiences that could lead to the observed effects. Some way to address this could be to correlate the frequency of touch-related events with any of the neural or behavioral outcomes. There are also many ways to modulate S1 activity (for example, whisker removal or ablation, or silencing S1 neurons) to test the author's circuit model.
- Similarly, immediate early gene activity is dynamic and is typically used to infer a snapshot of neuronal activity time-locked to a particular event (such as following a brief memory retrieval test). It is unclear what kinds of behaviors the control vs the enriched animals were experiencing prior to sacrificing for c-Fos analysis, besides inferring some kind of frequent tactile exposure in the enriched groups.
- Control mice in Figure 1 were socially isolated, and were thus deprived of both the enriched environment as well as social interaction, making it difficult to conclude if the cortical c-Fos enhancement is tied directly to purely tactile, or to social deficits. A further possibility is that social deprivation induces anxiety or depression-like effects that are reversed by enrichment, warranting a fundamentally different interpretation of the results. This could be addressed by using socially housed control mice instead.
- There are some discrepancies in the structural data in Figures 4 versus 7 that should be

mentioned. Tactile enrichment did not affect the number of branch points in Figure 7j, but did so in 4d

- In figure 4k,l, the basis of the "relative connectivity ratio" was not clear from the results narrative or figure legend. This should be explicitly stated. Assuming this means retrogradely labeled LEC neurons normalized to the activity-tagged starter population, it is also difficult to place this result in context with the other findings. Since there are differences in number (in some cases) and pattern of activated DG cells between control and enriched mice, it could simply be the paradigm does not remodel DG circuitry, as the authors conclude, but rather that there is differential recruitment of cells with pre-existing connectivity differences. This experiment does not discriminate between these possibilities.
- Figure 5 should be unpacked to explain precisely how this experiment was conducted and what is the meaning of direct versus indirect targeting of LEC neurons. How did the authors validate their anterograde Cre vector? Ultimately, it is unclear what has been established from this experiment that informs on the role of LEC-DG processing. For example, does convergent labeling of LEC neurons differ between control and enriched mice? Is the proportion of labeled L2 cells more than would be observed for other pathways in the brain (i.e. visual, olfactory, multimodal)? Further, it seems that most S1 inputs target deep layers of LEC.
- While it's easy to understand what a change in overall c-Fos counts implies (i.e changes in cellular activity of a region), I'm not sure what is implied by analyzing the distance between c-Fos+ cells. The authors call this a change in the activity pattern, yet what is the rationale to justify why this is a meaningful metric? Does this necessarily suggest they engage unique circuits? Some justification here would be appreciated.
- For DREADD manipulation of tagged populations of dDG and vDG, the authors use 4OHT-induced recombination on the last two days of 10-day enrichment. It is unclear to me why these time points were chosen or why, for the purpose of tagging, a brief period of tactile experience would not suffice given that ESARE levels will decay rapidly after activity. The authors also did not characterize the ESARE approach to confirm anticipated patterns of labeling (i.e. increased labeling in enriched versus control, clustered labeling in dDG) and, critically, did not quantify labeling in vehicle versus CNO condition. Finally, because they did not perform DREADD manipulations on control (nonenriched mice) it is not clear whether these behavioral effects depend on modulation of tactile enrichment activated cells or whether they could be elicited by stimulating any random population.

Minor comments:

- It would be nice to see some quantification of DCX/c-Fos data in Figure 2g. Also, I would suggest looking at different IEG, such as zif286, which is more likely to be seen in DCX+ cells compared to c-Fos or Arc.
- The cumulative distribution curves of the distance between c-Fos+ cells do not seem that meaningfully different from each other, despite its statistical significance (see Figure 2j & 3h).
- It's curious that behavioral effects were observed 1wk after the tactile enrichment paradigm, yet the c-Fos levels were observed to return to basal levels. I think this deserves some comment to explain.
- Figure 5 is rushed over pretty quickly in the text, I suggest elaborating on the results.
- While the authors claim the inputs onto "touch-tagged" neurons preferentially come from L2 of EC ("Taken together, tactile enrichment remodels DG neurons in an input-specific manner..."), it also appears that the control-tagged group has a similar preferential input from layer L2. Some kind of layer-specific quantification would be helpful to strengthen a claim that the "touch-tagged" population receives a layer-specific input, as opposed to simply strengthening the connectivity (this might be in supplemental figures, but I did not have access to the them).
- I'm unsure why there is a change in analysis for Figure 6h compared to 6d.

Title: Tactile Modulation of Memory and Anxiety Requires Dentate Granule Cells along the Dorsoventral Axis

Response to Reviewer 1: pages 1-6

Response to Reviewer 2: pages 7-17

Response to Reviewer 3: pages 18-30

Reviewer 1:

We thank the reviewer for the positive comments and valuable suggestions. Please find our detailed response below.

The manuscript from Wang and colleagues investigates the role of tactile modulation on memory and anxiety-like behaviour. It is a thorough, extensive, and multi-disciplinary study with many novel findings. The authors suggest that enriched tactile experience retunes the SI→LEC→dDG/vDG pathway and enhances DG neuron plasticity to modulate cognition and emotion.

This is a body of work potentially worthy of publication in such a high profile journal, but there are some short-comings. Limitations in methodological design limit the conclusions substantially. Additionally, further experiments are needed in order to fully to reveal the neural mechanisms of tactile enrichment on cognition.

1. The authors suggest that the data presented in figure 1c-f indicate that tactile input constitutes a major sensory source under MME, and raise the possibility of SI-associated neural circuits mediating the effects of MME on cognition and emotion. However, the data in Fig 1c-f is from multi-modal enrichment only, the authors should include in this figure (fig 1c-e) a tactile enrichment group so as to directly compare the effect of tactile experience from multi-modal experience on neuronal activation. (ie combine with supplementary figure 3e).

Furthermore, the authors should comment on the difference in magnitude between cfos

activation in S1 following MME (fig 1c) and TEE (supp fig 3e). The cfos activation in L4 appears less in the TEE group than the control of the MEE?

Response:

Data presented in previous Supplementary Fig. 3e were obtained from mice subjected to 20 days of tactile enrichment and multiple behavioral tests, whereas data shown in Fig. 1c-e were obtained from behavioral test-naïve mice with 10 days of multimodal enrichment. Therefore, it is inappropriate to combine these data. To address this issue as well as the first issue raised by Reviewer 2 and the third issue raised by Reviewer 3, we performed additional experiments to directly compare neuronal activation in both S1 and DG after tactile enrichment, social housing and multimodal enrichment. We found that multimodal enrichment increased neuronal activation in several regions of S1 (especially the barrel field and trunk regions), dorsal DG and ventral DG. Tactile enrichment also increased neuronal activation in the barrel field and trunk regions of S1 as well as ventral DG, and altered neuronal activation pattern in dorsal DG. These data are presented in revised **Fig. 2** and **Supplementary Fig. 4**. We also rephrased related statements in the results section (page 5, the last sentence of the 1st paragraph).

It should be mentioned that, based on reviewers' comments and the focus of this study, some figures (including previous Supplementary Fig. 3e) were either reorganized or removed from the revised manuscript.

2. It would be ideal to include an additional experimental group of artificial activation (chemogenetic or optogenetic) of S1 L4 on downstream activation of DG and LEC and compare to tactile induced activation.

Response:

Based on this suggestion and the first issue raised by Reviewer 3, we chose Scnn1a-Cre mice, in which Cre recombinase is expressed in S1 layer 4 excitatory neurons, and performed an

additional experiment to examine the modulation of LEC and DG activation by S1 inputs. We found that chemogenetic stimulation of S1 layer 4 neurons significantly increased the number of c-fos⁺ neurons in LEC, dorsal DG and ventral DG, and partly mimicked the influence of tactile enrichment on spatial patterns of c-fos⁺ neurons in DG. These data are presented in revised **Fig. 7a-e**.

3. The authors suggest that tactile inputs converge on the DG via LEC to shape cognitive process. Indeed, the authors showed that touch-tagged neurons in the DG received input from layer 2 LEC (Fig. 4j and Supplementary Fig. 7a). Thus, the manuscript would benefit from experimentally exploring whether inactivation of the LEC layer 2 during tactile enrichment blocks the downstream effects on DG activation and behaviour.

Response:

Currently, tools that specifically target LEC layer 2 neurons are unavailable. Based on the reviewer's suggestion, we performed an additional experiment in which the anterograde Cre-expressing AAV was injected to the S1, while the hM4Di-expressing AAV was injected to the LEC. In this way, the involvement of S1-innervated LEC neurons in tactile enrichment-evoked effects can be examined. The results show that chemogenetic inhibition of S1-innervated LEC neurons abolished the effects of tactile enrichment on object location discrimination and anxiety-related behavior. These data, which are presented in **Fig. 7f-i**, provide further evidence that the S1→LEC→DG pathway modulates the behavioral effects of tactile enrichment. Since these mice underwent behavioral testing and were killed 1 week after tactile enrichment ended when its effects on neuronal activation waned according to Fig. 3g, we did not compare DG activation between groups.

4. The authors persuasively show that their activity-dependent labeling method is indeed activity dependent. But they do not show that they have specificity for tactile stimulation. Would their interventions have the same behavioral effect if the stimulated home cage cells (labelled without tactile stimulation) or novel context engram cells? These are absolutely

crucial control experiments.

Response:

Indeed, a number of DG neurons express c-fos under basal conditions. We agree with the reviewer that a control experiment will strengthen our conclusion. To address this issue, we used chemogenetics to activate dDG or vDG neurons that were labeled under home-cage conditions. In contrast with TEE-tagged DG neurons, activation of control DG neurons did not influence spatial memory performance nor anxiety level. These data are shown in **Supplementary Fig. 11**.

5. How do the tactile enrichment effects on SI-LEC-DG plasticity compare to multimodal enrichment experience. The authors should include a MME group to the morphological analysis presented in figure 4 and 5.

Response:

Based on this suggestion as well as comments of Reviewer 3, we performed additional experiments and compared the impact of individual housing, social housing, tactile enrichment, and multimodal enrichment on the activation and structural plasticity of DG neurons. Morphological data are presented in **Supplementary Fig. 7b-i**. Compared with tactile enrichment that mainly increased mushroom spine density in DG granule cells, multimodal enrichment significantly increased thin and total spine density. No difference between control and group-housed mice was observed.

In the revised manuscript, several enrichment and housing conditions were included either to indicate the importance of tactile enrichment and introduce the major findings (multimodal enrichment), or to address the reviewers' concerns (multimodal enrichment, varied rows of bead curtain, and group housing). Because the current study primarily aims to examine the mechanisms underlying the behavioral effects of tactile enrichment, from Fig. 3 onwards we only focused on tactile enrichment and did not compare its effects with those of multimodal

enrichment or other enrichment/housing conditions for every experiment. We appreciate the reviewer's valuable suggestions and will explore these interesting questions in future studies.

6. What did the pattern of neuronal activation look like in the vDG (figure 2i and j)? Include a graph of vDG pattern of neuronal activation or discuss relative to dDG.

Response:

Based on the comments of Reviewer 2, we used nearest neighbor distance (Baddeley et al., 2015) instead of pairwise distance (referred to as “distance between c-fos⁺ cells” in the previous manuscript) to examine the spatial distribution pattern of activated dDG and vDG neurons. The procedure and rationale are described in the method section (page 34, 1st paragraph). Moreover, empty space distance maps were used in main figures to facilitate data visualization. For all related analyses, both dDG and vDG data are now presented (**Fig. 2**, **Fig. 3d-g**, and **Fig. 7d,e**).

7. The authors should include in supplementary figures the exploration times for the acquisition and test period for all the object recognition tasks.

Response:

We thank the reviewer for noting this issue. Considering the amount of object recognition data (18 sets of data in total) and space limitations, we calculated object preference ratio in the acquisition phase(s) instead of presenting absolute time spent exploring each object (otherwise the number of bars will be doubled for many data, or even quadrupled for object-in-place data). These data are summarized in **Supplementary Fig. 13** and **Supplementary Table 2**. For all these data, no difference was found among groups.

8. The authors should offer some insight as to why enrichment may affect both the dorsal and ventral DG SPB but only selectively affect the dorsal DG IPB (Fig 1e).

Response:

In some (but not all) experiments, we found differences in c-fos⁺ neuron density between the supra- and infra-pyramidal blades of DG. In the revised manuscript, we have added a few additional data (including **Fig. 2**, **Fig. 3d-g** and **Fig. 7d,e** for pattern analysis and several supplementary figures for density analysis). A more appropriate method (nearest neighbor distance) was applied to measure spatial patterns of activated DG neurons. In this case, it is more suitable to take the dentate gyrus as a whole. Therefore, we do not emphasize the difference between SPB and IPB in the revision.

9. Line 204: Authors should rephrase sentence. They are not restoring memories. They have restored ability to discriminate. This is an important distinction.

Response:

This sentence is now rephrased (page 13, 2nd paragraph, line 5-8).

10. Results section doesn't include any statistical information. It is in a table in the supplementary, however this is difficult to follow, can this be include in the text of results as is usually the case?

Response:

We prefer to include statistical information in the results section, which is more straightforward for readers. However, the revised manuscript contains 8 main figures and 13 supplementary figures. Including statistical information (especially for data analyzed by ANOVA or Kruskal-Wallis test followed by *post hoc* multiple comparisons) would greatly increase the length of the results section (currently already 10 pages in Word) and thus the whole manuscript. We thank the reviewer's suggestion and will continue to do so in our future work.

Reviewer 2:

We appreciate the reviewer's constructive comments and suggestions. Please find our detailed response below.

In the present manuscript Wang C. et al. addressed the important question of how tactile enrichment modulates memory and anxiety in mice. Towards this, the authors developed a novel model of tactile enrichment (TEE) and combined immunohistological, morphological and behavioral analyses with state-of-the art approaches such as rabies-based connectivity tracing and chemogenetic manipulation of neuronal activity. They showed that tactile enrichment reduced anxiety and improved memory. Interestingly, TEE-induced activation of cortical somatosensory (SI) and dentate gyrus (DG) granule neurons promoted synaptic plasticity in DG neurons and increased lateral entorhinal cortex (LEC) connections to experience-activated DG neurons. Moreover, the authors used an elegant approach to specifically modulate experience-activated neurons in the ventral and dorsal DG. These well-conducted experiments confirmed the role of TEE-activated granule neurons in the modulation of memory and anxiety.

Interestingly, the authors also showed that TEE is able to rescue memory deficits and ameliorates anxious behavior produced by early-life stress. This is an interesting and well-written paper which addresses important issues, but some concerns need to be addressed. Moreover, one of the described experiments seems to be impossible to have worked as the authors suggest.

1. One of the main issues of the manuscript concerns the characterization of the novel tactile enrichment model (TEE). Do the c-fos numbers correlate with the number of curtain zone entries? Does the half-curtain also activate c-fos? Moreover, can neuronal activation (c-fos+) be maximized by increasing the number of curtain rows (for example with 3 adjacent curtains)?

For curiosity, how specifically is this tactile circuit activating different sets of neurons, i.e are the same neurons activated every time the mice are stimulated? Is there a somatotopic

representation of activated neurons in S1?

Response:

To address these issues, some of which were also raised by other reviewers, we performed additional experiments and analyses.

First, we analyzed the relationship between activities in the curtain zone and the density of c-fos⁺ neurons in the S1 of mice housed in different cage setups (with or without bead curtain or nestlet). We found a significant correlation between the distance traveled in the curtain zone and c-fos⁺ neuron density in the thalamo-recipient layer 4 of S1 only in the Beads/Nestlet group (i.e., with the selected tactile enrichment setup). These data are shown in **Supplementary Fig. 2g**.

Second, we examined the influences of different rows of bead curtains (a half row, one row or three rows) on neuronal activation in the dorsal and ventral DG. A half row of curtain did not markedly increase the number of activated DG neurons. Compared to one row of curtain, three rows of curtain induced similar extent of neuronal activation in the ventral DG, but did not further increase c-fos⁺ neuron density in the dorsal or ventral DG. These data, which support that the selected one row of bead curtain is optimal, are shown in **Supplementary Fig. 3e,f**.

Third, to examine whether tactile experience enrichment (TEE)-tagged DG neurons are repeatedly activated under recurrent tactile enrichment, we labeled activated dDG neurons on different days of tactile enrichment (day 1, 5 or 10), and killed mice for c-fos immunostaining. Although the reactivation rate (i.e., c-fos⁺ cells among mCherry⁺ labeled cells) was not high (~3%, which is similar to some other studies, e.g., Wang et al., 2020), we did find that the reactivation rate of neurons labeled on the 5th day of tactile enrichment was higher than neurons labeled under home-cage conditions. Considering that c-fos immunostaining is not sensitive enough to better address this question, future studies using *in vivo* imaging

techniques are needed to reveal the dynamics of DG neuron activation/reactivation under tactile enrichment. These data are shown in **Supplementary Fig. 8e-h**. We also discussed this limitation in the revised manuscript (page 19, 2nd paragraph).

Fourth, the mouse S1 is a big structure that occupies ~18% of the neocortical surface area and includes 9 subregions, which makes cell counting across multiple subregions quite time-consuming. Using a deep learning approach (U-Net segmentation), we were able to quantify c-fos-immunostained neurons in layers 2-6 of five main S1 subregions whose boundaries are relatively clear (barrel field, forelimb, hindlimb, shoulder, and trunk regions). The results show that tactile enrichment mainly increased neuronal activation in S1 barrel field and trunk regions. Note that in this additional experiment, we also included a multimodal enrichment group and a social housing group to address other reviewers' concerns. These data are presented in **Supplementary Fig. 4**.

2. A serious technical issue concerns the rabies-tracing experiments. What is the rationale for injecting the rabies 2 weeks before 4-OHT? In principle, TVA expression (here, dependent on 4-OHT-induced activation of Cre) is required to render the cells susceptible to infection by the rabies. Thus, this referee thinks that the experiment as described cannot work as the rabies virus should remain in place for 2 weeks before being able to infect cells.

Also, efficient tracing requires (after 4-OHT injection) both recombination and spread of the rabies to occur; From the referee's experience 2 days would not be enough for that.

Response:

From the experimental design perspective, the rabies virus (RABV) can be delivered to the DG before, during or after the 10-day tactile enrichment. Because stereotaxic microinjection of viral vectors is an invasive approach, it is not ideal to deliver RABV during tactile enrichment. RABV can also be delivered immediately after tactile enrichment, but the injection procedure may influence synaptic plasticity in the regions of interest and a few days are required for trans-synaptic tracing, during which the effect of tactile enrichment on

synaptic connectivity likely wanes. Taking these factors into consideration, we decided to inject RABV before tactile enrichment started, and made sure that mice could fully recover from surgery and viral injection.

Indeed, as the reviewer pointed out, before 4-OHT administration, the RABV-EnvA- Δ G-DsRed we used was unable to infect neurons without TVA expression, and thus no viral replication occurred in target neurons. According to the literature, unlike many other neurotropic viruses, RABV can minimize tissue inflammation, escape the host immune response and evade antiviral immune clearance from the brain (Davis et al., 2015; Hooper et al., 2011; Lafon, 2011). The lack of glycoprotein, a molecular determinant of pathogenicity, may also enable this modified RABV to escape immune recognition. In addition, our approach did not affect the integrity of the blood–brain barrier, which limits the entry of peripheral immune cells to the brain. Considering that wild-type RABV can survive in the salivary glands and saliva for weeks, the RABV we injected could technically remain and survive in the injection sites (i.e., DG) for days.

Concerning the interval between 4-OHT injection and killing, we apologize for not providing sufficient details in the previous manuscript, which brought confusion. For results shown in Fig. 4h-1 and Fig. 5, the tactile enrichment procedure started at 1 hour before the dark phase on Day 16. The first 4-OHT injection was performed at 1 hour before the dark phase on Day 24 (i.e., the beginning of the 9th day of enrichment), and the second was performed 24 hours later (the beginning of the last day of enrichment). Mice were transferred to standard cages at 1 hour before the dark phase on Day 26, and were killed ~48 hours later. Therefore, the interval between the first 4-OHT administration and killing was 4 instead of 2 days. In the revised manuscript, we have clearly specified these details (e.g., page 28, 1st paragraph).

We set this survival time based on our pilot experiment and previous studies. In the pilot experiment, we first chose a more commonly used interval between the start of RABV replication and killing (i.e., 7 days), but 2 mice died 5-6 days after the first 4-OHT injection

(possibly due to the cytotoxicity of RABV). Therefore, we shortened this interval to 4 days based on previous reports (Mori and Morimoto, 2014; Osanai et al., 2017) to avoid potential mortality (Lavin et al., 2019). It should also be noted that, after subcutaneous 4-OHT injection, 4-OHT can rapidly enter the brain (Jahn et al., 2018; Turner et al., 2011). In addition, the half-life of ER^{T2}CreER^{T2}-PEST is 2.9 hours (Kawashima et al., 2013). Upon 4-OHT injection, the helper AAVs and AAV-ESARE-ER^{T2}CreER^{T2} that were injected 3 weeks before had sufficiently replicated in infected neurons, and ER^{T2}CreER^{T2} was already available in the cytoplasm of neuronal populations that were activated under tactile enrichment about 3 hours before. In this case, 4-OHT could induce gene recombination and thus the expression of TVA and glycoprotein within hours.

In addition to the data shown in Fig. 4 and Fig. 5, another unpublished study from our group supports that such survival time works. In this experiment, we injected helper AAVs and RABV (using the same batch of RABV and the same injection volume with those of the current study) to the hypothalamic paraventricular nucleus (PVN) of AVP-Cre mice (with Cre recombinase expressed in arginine vasopressin neurons). At 3.5 days after RABV injection, mice were killed and brains processed. As shown below, neurons in several hypothalamic nuclei (e.g. arcuate hypothalamic nucleus, Arc) and some distant brain regions (e.g. ventral subiculum, VS) that innervate PVN AVP neurons can be labeled.

Taken together, with our experimental design and injection strategies, LEC neurons that innervate TEE-tagged DG neurons can be labeled.

3. Also, there are issues with the experimental approach used to trace S1-to-LEC connections. If the AAV injected in S1 is indeed anterogradely transported, labelled axon terminals (rather than green somas) would be expected in LEC.

Response:

Axon-like structures (either *en passant* or terminal) can be observed in the LEC in the original images (e.g., Fig. 5d, the right panel). However, the fluorescence intensity of neuronal soma is much higher compared to these axon-like structures in the LEC. In addition, these axon-like structures that theoretically originate from S1 are intertwined with neurites of infected LEC neurons, which further mask them in the representative images. In the revised manuscript, we tried to reduce the background noise and optimized the brightness/contrast of representative images in **Fig. 5c,d** and **Supplementary Fig. 9b,c** to better reveal these structures.

We also performed an additional experiment in which the anterograde Cre-expressing AAV was injected to the S1 of C57BL/6 mice, while the Cre-dependent AAV that expressed hM4Di and mCherry was injected to the LEC. As shown in revised **Fig. 7g**, S1-innervated, mCherry-expressing LEC neurons can be clearly seen, which provides additional evidence that the anterograde AAV is working in our hands. It should also be mentioned that the same AAV (from the same company) has been successfully used in some other studies as well (e.g., Huang et al., 2019).

4. The quality of the images in Fig S1, S3E and G, S4, S6C and S8D need improvements. For all supplementary figures, higher resolution pictures would be required. The images shown in Fig 6C and G are insets of Fig S7F and G, thus should not be processed differently.

Response:

We have modified **Supplementary Figs. 1**, 3g (now **5b** in the revised manuscript), 6c (**8c** in the revised manuscript), and 8d (**10d** in the revised manuscript) either by optimizing

brightness and contrast or providing new images with better resolution.

Supplementary Fig. 7f,g (**9f,g** in the revised manuscript) were captured with a 10× objective to present an overview of the whole DG, whereas **Fig. 6c** and **Fig. 6g** were imaged at a higher magnification with a 40× objective to better reveal the details of mCherry⁺ and c-fos⁺ neurons. Since these two sets of images were obtained using different parameters, they were processed differently. To avoid potential misunderstanding, we always specify whether an image in the inset was digitally magnified or captured at a higher magnification in the revised figure legends and Supplementary Information (e.g., page 49, 2nd paragraph, line 5).

It should be mentioned that, based on reviewers' comments and the focus of this study, some figures (including previous Supplementary Fig. 3e) were either reorganized or removed from the revised manuscript.

5. The authors should indicate precisely what is depicted in the correlation analysis (Fig 1F).

Response:

Considering all reviewers' comments on Figs. 1 and 3, we performed additional experiments and made substantial revisions accordingly. To make the main message in Figs. 1-3 more focused and straightforward, we removed correlation analyses that were presented in former Fig. 1f and Supplementary Fig. 4c from the revised manuscript.

6. Was the morphometric analysis performed on activated neurons (E-SARE labelled DG-neurons)? This would be useful to specify.

Response:

For revised **Fig. 4a-g** and **Fig. 8h-m** as well as **Supplementary Figs. 7** and **12d-g**, we used the Golgi-Cox method to examine dendrite and spine morphology. For revised **Supplementary Fig. 6a-h**, we used EGFP-expressing retrovirus to label adult-born DG

neurons. We specify these details in the revised results and methods sections (e.g., page 9, 1st paragraph; page 14, 2nd paragraph; page 35, 2nd paragraph).

7. From Fig S1 to conclude, the increase in c-fos positive cell number in S1 seem to be generalized to all layer (and not restricted to L4). Similarly for in V1 in Fig S1C, the increase seem to affect all layers.

Also, in Fig S3E, it is not possible to appreciate the increase in neuronal activation in L2-4 in S1.

Response:

For data shown in Fig. 1d,e and Supplementary Fig. 1, there were indeed trends that the density of neurons in S1 layer 6 and V1 layers 4 and 6 was increased in the multimodal enrichment group. However, these visual differences were not statistically significant (S1 L6, $P = 0.082$; V1 L4, $P = 0.108$; V1 L6, $P = 0.099$). In the revision, we performed c-fos immunohistochemistry that yields better signal-to-noise ratio than immunofluorescence, included more subregions of S1, and compared the effects of tactile enrichment, social housing and multimodal enrichment to better reveal the differences among S1 subregions and layers (**Supplementary Fig. S4**). These data reveal that multimodal enrichment increased neuronal activation in layers 2-4 of the barrel field and trunk regions and deep layers of all examined S1 regions.

For previous Fig. 2g-j and Supplementary Fig. 3e, because these mice underwent both tactile enrichment and behavior test battery, the results about neuronal activation in DG and S1 may be confounded by the carry-over effects of behavioral testing. Therefore, we replaced these data by new data from behavioral test-naïve mice in the revised manuscript.

8. The meaning of the pattern of activation analysis (Fig 2I and J) should be better explained. Also, in the clustering analysis, the distance-path should follow the dentate gyrus shape. Instead Fig 2J depicts a straight line crossing the hilus.

Response:

Based on the reviewer's suggestion, we chose a more appropriate and reliable method (nearest neighbor distance), instead of pairwise distance (referred to as "distance between c-fos⁺ cells" in the previous manuscript) that made many interpoint comparisons cross the polymorphic layer, to analyze spatial point pattern (Baddeley et al., 2015). As this method allows to analyze point pattern in complex shapes, we also included the infrapyramidal blade and took the dentate gyrus as a whole in the revised manuscript. The rationale and procedures of activation pattern analysis are now detailed in the method section (page 34, 1st paragraph). New data are presented in **Fig. 2**, **Fig. 3d-g** and **Fig. 7d,e**.

9. A proper analysis of adult-born neurons maturation (Fig S3H) would require thymidine-analogue experiments.

Response:

Thymidine analogues are primarily used to examine cell proliferation, differentiation and survival (reviewed by Deng et al., 2010; Ming and Song, 2005). In this study, we already used MCM2 and doublecortin to quantify cell proliferation and differentiation respectively (revised **Supplementary Fig. 5a,b**). Based on the reviewer's suggestion, we labeled adult-born dentate granule cells with EGFP-expressing retrovirus, and evaluated the effects of tactile enrichment on the morphological maturation of newly generated DG neurons (Zhao et al., 2006). The results, which are presented in **Supplementary Fig. 6a-h**, show that tactile enrichment did not affect the morphological maturation of dendrites and spines in newborn granule cells.

10. Fig S4E y-axis should indicate that it represents the "Percentage of calbindin positive neurons among activated cells".

Response:

Considering the focus of this manuscript, we replaced this figure that showed the effects of

short-term tactile enrichment by **Supplementary Fig. S6i,j**, which showed the effects of continuous tactile enrichment. The Y-axis has been modified according to the reviewer's suggestion.

11. In Fig 5E, how did the authors discriminate between direct and indirect connections? Do yellow cells correspond to direct connections? How did the authors label the indirect ones?

Response:

As reported and validated previously (Zhou et al., 2019; Zingg et al., 2017), the anterograde AAV2/1-hSyn-Cre vector and the retrograde RABV system we used mediate monosynaptic tracing. Therefore, LEC neurons labeled by both mCherry (retrogradely labeled from TEE-tagged DG neurons) and EGFP (induced by the Cre recombinase that was anterogradely transported from S1), which yielded a yellow appearance, were directly connected with S1 and DG. Such connection among S1, LEC and DG are considered as direct connections. LEC neurons that only expressed EGFP were directly innervated by S1, but did not directly project to tactile experience-activated neurons. Such connections are considered as indirect connections. We apologize for the brevity of the previous results section, and have provided more details in the revised manuscript (page 10, 2nd paragraph).

12. How many sections and cells were analyzed for IHC quantifications?

Response:

Each region of interest was measured bilaterally in at least three sections. For some figures like Supplementary Fig. 4, up to 12 sections were chosen from each mouse and the whole extent of the subregion on the section (e.g., barrel field) was measured bilaterally. These details are stated in the revised methods section (e.g., page 33, 1st and 2nd paragraphs; page 34, 1st paragraph).

13. Could the authors explain why they used only males in their study, excepted for the TEE

model establishment where they used only females?

Response:

As one of the first experiments of this study, the model establishment experiment aimed to optimize the setup of the tactile enrichment environment. Since sex difference is unlikely to play a role, we used 25 female mice that were available then to minimize the use of animals. These females did not undergo cognitive or anxiety tests, and were killed after exploration of specific cages. Considering this experiment does not contribute to the main conclusions and we do not focus on sex difference in this manuscript, only male mice were used for all main experiments, which is a commonly used strategy. In the revised discussion section, we acknowledged this limitation and pointed out that sex should be considered in future studies (page 20, 1st paragraph).

Reviewer 3:

We thank the reviewer for the constructive comments and suggestions. Please find our detailed response below.

Wang et al. propose an intriguing role of prolonged tactile stimulation with cognitive enhancements and anxiolytic behavior. The authors demonstrate that 1) tactile stimulation causes structural and activity modifications in the dentate gyrus (DG), accompanied by increased lateral entorhinal cortex connections to touch-associated DG neurons 2) manipulating activity of the touch-associated neurons bi-directionally modifies the behavior and anxiety effects, and 3) tactile stimulation reverses some of memory deficits and anxiety-like phenotypes associated with early life stress. The experiments are well designed, and the major strengths of this paper are its novelty and combination of powerfully selective technical approaches. Unfortunately, however, the study suffers from the lack of critical controls. Ultimately, the key conclusions that tactile enrichment strengthens an SI->LEC->DG circuit mediating recruitment of DG cells, and that DG cells activated by enrichment selectively control behavior, requires overinterpretation of the findings. My major and minor comments are detailed below.

1. It is difficult to conclusively say that the increased c-Fos related to the multi- or unimodal tactile stimulation paradigms is associated with the touch sensation per se, or to a variety of other behaviors that could be related to interacting with the objects (e.g. increases in locomotion or exercise, positive affective associations to the objects, experiencing unique “contexts” generated by segmenting the cage into separate zones). While this does not necessarily undermine the results of the paper, it does muddy the water on identifying the kinds of sensory experiences that could lead to the observed effects. Some way to address this could be to correlate the frequency of touch-related events with any of the neural or behavioral outcomes. There are also many ways to modulate SI activity (for example, whisker removal or ablation, or silencing SI neurons) to test the author’s circuit model.

Response:

Some previously presented data (in Fig. 2g-j and Supplementary Fig. 3e) were obtained from mice that underwent both tactile enrichment and behavior testing, which are now replaced by new data obtained from behavioral test-naïve mice. We also analyzed the relationship between activities in the curtain zone and c-fos⁺ neuron density in S1. We found that the distance traveled in the curtain zone significantly correlated with the number of activated neurons in S1 layer 4, which receives tactile information from the ventral posteromedial thalamus, only in the Beads/Nestlet group (i.e., with the selected tactile enrichment setup). These data are shown in **Supplementary Fig. 2g**.

We used Scnn1a-Cre mice that express Cre recombinase in S1 layer 4 excitatory neurons to examine the modulation of LEC and DG activation by S1 activity. We found that chemogenetic stimulation of S1 layer 4 neurons increased the number of c-fos⁺ neurons in LEC, dorsal DG and ventral DG, and partly mimicked the influence of tactile enrichment on spatial patterns of c-fos⁺ neurons in DG. These data are presented in **Fig. 7a-e**.

We also performed an additional experiment, in which the anterograde Cre-expressing AAV was injected to the S1, while the Cre-dependent, hM4Di-expressing AAV was injected to the LEC. With this experimental design, the role of S1-innervated LEC neurons in tactile enrichment-evoked effects can be examined. The results show that chemogenetic inhibition of S1-innervated LEC neurons abolished the effects of tactile enrichment on object location discrimination and anxiety-related behavior. These data are presented in **Fig. 7f-i**.

Taken together, these additional data support our hypothesis that the S1→LEC→DG pathway modulates the behavioral effects of tactile enrichment.

2. Similarly, immediate early gene activity is dynamic and is typically used to infer a snapshot of neuronal activity time-locked to a particular event (such as following a brief memory retrieval test). It is unclear what kinds of behaviors the control vs the enriched

animals were experiencing prior to sacrificing for c-Fos analysis, besides inferring some kind of frequent tactile exposure in the enriched groups.

Response:

As mentioned above, some data presented in our previous manuscript were obtained from mice underwent both tactile enrichment and behavioral testing. Therefore, neuronal activation in DG might be the mixed results of both factors, which makes data interpretation non-straightforward. In the revised manuscript, we have removed these data and performed additional experiments to make sure that all c-fos or Egr1 staining were done in testing-naïve animals. These new data are presented in **Fig. 2, Supplementary Fig. 4 and Supplementary Fig. 6i,j.**

3. Control mice in Figure 1 were socially isolated, and were thus deprived of both the enriched environment as well as social interaction, making it difficult to conclude if the cortical c-Fos enhancement is tied directly to purely tactile, or to social deficits. A further possibility is that social deprivation induces anxiety or depression-like effects that are reversed by enrichment, warranting a fundamentally different interpretation of the results. This could be addressed by using socially housed control mice instead.

Response:

In the current study, adult mice remained group-housed until at 1 week before tactile enrichment or surgery. This is specified in the methods section (e.g., page 21, 1st and 3rd paragraphs; page 26, 2nd paragraph). In addition, in contrast to rats where isolation housing is a powerful stressor, previous studies show that single housing of male mice is not a stressful condition, and adult male mice are usually individually housed to avoid excessive fighting and thus injury (Arndt et al., 2009; Friedman et al., 2017; LeGates et al., 2012; just to list a few). We acknowledge that our previous experiments did not take the housing condition into consideration. To exclude the possibility that individual housing alters basal cognitive performance and anxiety level, we compared the effects of single housing and group housing

on object location memory and anxiety-related behavior. As shown in **Supplementary Fig. 3g,h**, no difference was found between individually housed mice (CTL) and socially housed mice (SOC). Moreover, based on the reviewer's comments and the first issue raised by Reviewer 1, we also compared the effects of individual versus social housing on neuronal activation in S1 and DG and the structural plasticity of DG granule cells (**Fig. 2, Supplementary Figs. 4 and 7**). We found that these two groups in general did not differ in the activation and morphology of DG neurons, although socially housed mice had more activated neurons in layers 2-4 of S1 barrel field than individually housed mice. Taken together, our new data suggest that individual housing does not alter cognitive performance, anxiety level nor DG neuron plasticity compared to group-housed mice. We therefore keep individually housed mice as the controls.

4. There are some discrepancies in the structural data in Figures 4 versus 7 that should be mentioned. Tactile enrichment did not affect the number of branch points in Figure 7j, but did so in 4d

Response:

We thank the reviewer for noting this issue. For dendrite morphology analysis, total length and the number of intersections at concentric circles centered around the soma are more frequently used parameters than the number of branch points, and these parameters did not always change in the same direction (e.g., Gallitano et al., 2016; Shansky et al., 2009). In Fig. 4d, although we observed a difference in branch point number between groups, the effects were subtle and not seen in data shown in Fig. 8j and Supplementary Fig. 7f (additional data obtained during revision). Considering that no difference in dendritic length or the number of intersections was found, we concluded that tactile enrichment did not markedly affect dendritic complexity. These discrepancies were discussed in the revised discussion section (page 17, the last 3 sentences).

5. In figure 4k,l, the basis of the “relative connectivity ratio” was not clear from the results

narrative or figure legend. This should be explicitly stated. Assuming this means retrogradely labeled LEC neurons normalized to the activity-tagged starter population, it is also difficult to place this result in context with the other findings. Since there are differences in number (in some cases) and pattern of activated DG cells between control and enriched mice, it could simply be the paradigm does not remodel DG circuitry, as the authors conclude, but rather that there is differential recruitment of cells with pre-existing connectivity differences. This experiment does not discriminate between these possibilities.

Response:

We apologize for not clearly stating the rationale of analyzing connectivity ratio in the previous manuscript. “Presynaptic connectivity ratio”, which is calculated as the ratio of presynaptic input cell number to starter cell number, has been used in many recent studies to reflect the strength of connectivity between rabies virus (RABV)-retrogradely labeled neurons and starter neurons (e.g., Deshpande et al., 2013; McAvoy et al., 2016; Skelton et al., 2019; Terreros-Roncal et al., 2019). In the revised manuscript, we specified the rationale and cited related papers in the methods section (page 32, 2nd paragraph, the last 4 lines), and stated more clearly in the figure legend (page 48, the last 3 lines).

Due to technical limitations (e.g., variations in the exact site of viral delivery and spreading of virus), only a fraction of TEE-activated DG neurons could be labeled and the labeling efficiency of AAV and RABV varied among animals. As pointed out by the reviewer, a minor difference in the number of labeled starter cells in vDG was seen between the control and tactile enrichment groups (Fig. 4k). However, in comparison with several previous studies (e.g., Skelton et al., 2019; Terreros-Roncal et al., 2019) in which the connectivity ratio was analyzed even though the number of starter cells was significantly different between groups, no significant statistical difference was found in this experiment. Therefore, the “relative connectivity ratio” used here, which takes both the number of starter cells and the number of retrogradely labeled cells into consideration (instead of merely counting retrogradely labeled cells), can reveal the structural plasticity of the LEC-DG circuit to a certain extent. In

addition, because the only between-group factor was with or without enriched tactile experience, such experimental design minimized potential differences in pre-existing connectivity of labeled DG neurons between groups. Nonetheless, we agree with the reviewer that the meaning of these structural data is limited and evidence at other levels (e.g., electrophysiology) would strengthen our conclusion. In the revised manuscript, we acknowledged and discussed this limitation (page 19, 2nd paragraph).

6. Figure 5 should be unpacked to explain precisely how this experiment was conducted and what is the meaning of direct versus indirect targeting of LEC neurons. How did the authors validate their anterograde Cre vector? Ultimately, it is unclear what has been established from this experiment that informs on the role of LEC-DG processing. For example, does convergent labeling of LEC neurons differ between control and enriched mice? Is the proportion of labeled L2 cells more than would be observed for other pathways in the brain (i.e. visual, olfactory, multimodal)? Further, it seems that most S1 inputs target deep layers of LEC.

Response:

We have revised this part and more details for Fig. 5 are now provided in the results section (page 10, 2nd paragraph). To address the reviewer's concern about the anterograde AAV2/1-hSyn-Cre as well as the third issue raised by Reviewer 1 and the third issue raised by Reviewer 2, we performed an additional experiment in which the anterograde AAV was injected to the S1 of C57BL/6 mice, and the Cre-dependent AAV that expressed hM4Di and mCherry was injected to the LEC. As presented in revised **Fig. 7g**, mCherry-expressing LEC neurons can be clearly seen, which shows that the anterograde AAV is working in our hands. Note that the same AAV (from the same company) has been successfully used in some other studies as well (e.g., Huang et al., 2019).

The entorhinal cortex, which comprises the lateral and medial subdivisions in mice, provides the major input to the dentate gyrus via the perforant path. It has been established by many

studies that reelin-expressing LEC layer 2 cells project to DG, while a subset of neurons in the deep layers of LEC (mainly layer 5b) target DG-projecting LEC layer 2 cells (Nilssen et al., 2019). The presentation and interpretation of data in Fig. 5 were based on this established connectivity. As the reviewer pointed out, S1 mainly innervated neurons in LEC deep layers. These results suggest that most inputs from S1 may be firstly processed by neurons in LEC deep layers before reaching DG. However, to examine how S1 inputs are processed and integrated in the LEC requires further anatomical, electrophysiological, and molecular experiments and is beyond the scope of the current manuscript. In addition, even in the tactile enrichment group, only a very small number of LEC layer 2 cells directly bridged S1 and DG (less than 0.4% of S1-innervated LEC neurons). Therefore, we did not include a control group and make between-group comparisons on these convergently labeled neurons that were sparse. We have rewritten the results and discussion to make the main message clearer (page 10, 2nd paragraph and page 18, 2nd paragraph).

While the current study focuses on the S1→LEC→DG pathway, we agree with the reviewer that it would be interesting to investigate inputs from other primary sensory cortices to LEC→DG and compare their roles in cognition and anxiety. We mentioned this future direction in the discussion section (page 19, 2nd paragraph).

7. While it's easy to understand what a change in overall c-Fos counts implies (i.e changes in cellular activity of a region), I'm not sure what is implied by analyzing the distance between c-Fos+ cells. The authors call this a change in the activity pattern, yet what is the rationale to justify why this is a meaningful metric? Does this necessarily suggest they engage unique circuits? Some justification here would be appreciated.

Response:

Pairwise distance (referred to as “distance between c-fos⁺ cells” in the previous manuscript), nearest neighbor distance, and empty space distance are classical methods to analyze the spatial distribution pattern of points (c-fos⁺ neurons in this case) and interpoint interaction

(Baddeley et al., 2015). Although these analyses do not give information about the connectivity or molecular profile of activated neurons, different spatial patterns between or among groups provide a basis for further examination of the mechanisms underlying such difference. We apologize for not making this point clear previously. In the revised manuscript, we specified the rationale (page 34, 1st paragraph) and rephrased the description in the results section (e.g., page 6-8). Please also see our response to the 10th issue below.

8. For DREADD manipulation of tagged populations of dDG and vDG, the authors use 4OHT-induced recombination on the last two days of 10-day enrichment. It is unclear to me why these time points were chosen or why, for the purpose of tagging, a brief period of tactile experience would not suffice given that ESARE levels will decay rapidly after activity. The authors also did not characterize the ESARE approach to confirm anticipated patterns of labeling (i.e. increased labeling in enriched versus control, clustered labeling in dDG) and, critically, did not quantify labeling in vehicle versus CNO condition. Finally, because they did not perform DREADD manipulations on control (nonenriched mice) it is not clear whether these behavioral effects depend on modulation of tactile enrichment activated cells or whether they could be elicited by stimulating any random population.

Response:

Based on the original publication (Kawashima et al., 2013) and our experience, E-SARE is sensitive to sensory stimulation. For example, a 10-min light stimulation (Kawashima et al.) or a 15-min whisker stimulation (Supplementary Fig. 8a-d) can efficiently activate E-SARE. An *in vivo* imaging study in mice using two-photon microscopy also provides evidence that E-SARE activation in CA1 pyramidal neurons peaks at 6-8 hours after exploration of an enriched environment for 2 hours (Attardo et al., 2018). Therefore, the response of E-SARE is rapid and its activity can last a few hours in mice with continuously enriched experience. In the present study, although each visit to the bead curtain zone was brief, mice generally explored the curtain thousands of times per day (estimated based on Supplementary Fig. 2e). Such repeated explorations of the bead curtain for 10 days could evoke recurrent activation of

related neurons, and thus maintain E-SARE expression in subpopulations of DG neurons. In addition, the last 2 days of tactile enrichment were selected for neuronal labeling according to previous findings that the activation of E-SARE stabilizes after repeated exposure to the same enriched environment (Attardo et al., 2018). We specified this rationale in the revised manuscript (page 27, the last paragraph and page 28, 1st paragraph).

Besides the E-SARE virus, 4-hydroxytamoxifen (4-OHT) injection strategies should also be taken into consideration. In our neuron labeling experiments, 4-OHT (dissolved in oil) was injected subcutaneously at 1 hour before the dark phase (active phase for mice) on specific days during tactile enrichment. As revealed by a recent study (Jahn et al., 2018), after a single intraperitoneal injection of tamoxifen (TAM), the level of its main bioactive metabolite 4-OHT (oxidized from TAM in the liver) in the brain peaks at 8 hours after injection. The half-life of 4-OHT in this case is ~16 hr, which can be extended to >20 hr after multiple TAM injections. In addition, subcutaneously injected substances are generally absorbed at a slower rate compared with the intraperitoneal route, which provides a more sustained effect (Turner et al., 2011). Based on the above-mentioned evidence, our 4-OHT injection strategies (label TEE-activated neurons on days 9-10 of the enrichment procedure for most experiments and days 6-10 for a validation experiment) are feasible and could make 4-OHT available in the brain at a functional level, thus opening up a time window for neuron labeling.

The pattern of labeled DG neurons by AAV-ESARE-ER^{T2}CreER^{T2} not only requires increased activity in neuronal ensembles, but also heavily depends on the range of viral infection. Technically, it is difficult to spread AAV to the whole dDG or vDG by stereotaxic microinjection, and variations in labeling among animals are inevitable. In this case, spatial pattern analysis is not applicable. For instance, if a small region of SPB or IPB is not infected by the virus and thus no neuron is labeled, this will make labeled neurons spatially much closer. Based on the reviewer's comments, we quantified the density of labeled neurons for data presented in Fig. 7, which are now shown in **Supplementary Fig. 9h,i**. For other data, we have checked each brain section from each animal to make sure that the extent of viral

infection (judging by mCherry immunofluorescence) in the target region was correct.

We also performed an additional experiment to address the reviewer's concerns as well as the first issue raised by Reviewer 2. In this experiment, we quantified the number of neurons that were labeled under home-cage conditions (without tactile enrichment) or a specific day during tactile enrichment (day 1, 5, or 10). The results show that the density of labeled neurons was comparable among groups, and that the reactivation rate of neurons labeled on the 5th day of tactile enrichment was higher than neurons labeled under home-cage conditions. These data are now presented in **Supplementary Fig. 8e-h**.

Moreover, we performed an important control experiment as suggested by the reviewer. In this experiment, mice remained housed under standard conditions, and dDG or vDG neurons that were active under basal conditions were labeled and later manipulated. The results show that chemogenetic activation of these control dDG or vDG neurons did not significantly affect spatial memory performance nor anxiety-related behavior. These data are presented in **Supplementary Fig. 11**.

9. It would be nice to see some quantification of DCX/c-Fos data in Figure 2g. Also, I would suggest looking at different IEG, such as zif286, which is more likely to be seen in DCX+ cells compared to c-Fos or Arc.

Response:

Consistent with the reviewer's comments, the colocalization between doublecortin and c-fos in the dentate gyrus was rarely observed in this experiment, which was mentioned in the previous results section. Among all samples examined (6 mice per group and 8 sections per mouse), we only found 2 doublecortin/c-fos double-labeled cells in the control group (2 mice with 1 double-labeled vDG granule cell each). No doublecortin/c-fos double-labeled cell was seen in the tactile enrichment group. Therefore, statistical analysis could not be applied and results were not presented previously.

According to the reviewer's suggestion, we performed an additional experiment using early growth response 1 (Egr1, also known as zif268) to label activated neurons. As shown in **Supplementary Fig. 6i**, about 1% of Egr1⁺ activated cells expressed doublecortin. However, there was no difference between the control group and the tactile enrichment group.

10. The cumulative distribution curves of the distance between c-Fos+ cells do not seem that meaningfully different from each other, despite its statistical significance (see Figure 2j & 3h).

Response:

Because additional experiments were performed and new data have been added to the revised manuscript, we included both the supra- and infra-pyramidal blades and take the two subregions as a whole. In this case, it is not suitable to use the pairwise distance (i.e., distance between c-fos⁺ cells), which crosses the hilus quite frequently (as Reviewer 2 pointed out), to reflect the activation pattern. Instead, we now use the nearest neighbor distance, a reliable and more suitable method to analyze spatial point pattern (Baddeley et al., 2015). In addition, empty space distance maps were used in main figures to facilitate data visualization. The rationale and procedures of pattern analysis are detailed in the method section (page 34, 1st paragraph). The revised data are shown in **Fig. 2**, **Fig. 3d-g** and **Fig. 7d,e**, but are presented differently (scatter plot) from the previous data.

11. It's curious that behavioral effects were observed 1wk after the tactile enrichment paradigm, yet the c-Fos levels were observed to return to basal levels. I think this deserves some comment to explain.

Response:

Indeed, the activation pattern of neurons in DG can not explain behavioral data at 1 week after tactile enrichment, which indicates that lasting changes in these neurons may be needed to modulate the behavioral effects of tactile enrichment. As suggested by environmental

enrichment studies, experience-induced lasting molecular and structural changes may be responsible for the temporal effects of tactile enrichment. For example, environmental enrichment-induced changes in dendritic spine formation and elimination may last for weeks to months (Yang et al., 2009). Environmental enrichment also leads to genome-wide alterations in gene transcription (e.g., Vallès et al., 2011). Therefore, tactile enrichment-evoked changes at molecular and structural levels may contribute to the observed effects on memory and anxiety, which is an interesting topic for future studies. We have revised the results (page 8, 1st paragraph) and discussion (page 16, lasted paragraph and page 17, 1st paragraph), and pointed out potential directions for future studies (page 19, last paragraph).

12. Figure 5 is rushed over pretty quickly in the text, I suggest elaborating on the results.

Response:

We revised this part and now more information is provided to the results section (page 10, 2nd paragraph).

13. While the authors claim the inputs onto “touch-tagged” neurons preferentially come from L2 of EC (“Taken together, tactile enrichment remodels DG neurons in an input-specific manner...”), it also appears that the control-tagged group has a similar preferential input from layer L2. Some kind of layer-specific quantification would be helpful to strengthen a claim that the “touch-tagged” population receives a layer-specific input, as opposed to simply strengthening the connectivity (this might be in supplemental figures, but I did not have access to the them).

Response:

In agreement with the reviewer’s comments and the literature, we found that dentate granule cells, regardless of control or TEE-tagged, are mostly innervated by LEC layer 2 cells (presented in **Fig. 4j-l**). Thus, it is not justifiable to claim that the structure and connectivity

modifications in TEE-tagged DG neurons are input- or layer-specific. We have removed such claims from the revised manuscript. Moreover, together with new data obtained during revision, we have rewritten the results and discussion accordingly.

14. I'm unsure why there is a change in analysis for Figure 6h compared to 6d.

Response:

For this dataset, we firstly analyzed the potential difference in discrimination index among groups using one-way ANOVA, which was applied for other similar datasets. However, due to within-group variation (especially in the dDG-CNO group), no significant statistical difference was found among groups. We then used an alternative approach to examine whether each group of mice could distinguish the two objects by comparing the relative time spent exploring the relocated object versus the stationary object, which revealed that mice with chemogenetic inhibition of TEE-tagged dDG neurons indeed failed to discriminate the objects. This is a valid approach that was used in our previous work (e.g., Li et al., 2017) as well as other studies (discussed in Akkerman et al., 2012; Bevins and Besheer, 2006) to evaluate cognitive performance in object recognition tasks.

References:

- Akkerman, S., Blokland, A., Reneerkens, O., van Goethem, N.P., Bollen, E., Gijssels, H.J., *et al.* (2012). Object recognition testing: methodological considerations on exploration and discrimination measures. *Behav. Brain Res.* 232, 335-347.
- Arndt, S.S., Laarakker, M.C., van Lith, H.A., van der Staay, F.J., Gieling, E., Salomons, A.R., *et al.* (2009). Individual housing of mice--impact on behaviour and stress responses. *Physiol. Behav.* 97, 385-393.
- Attardo, A., Lu, J., Kawashima, T., Okuno, H., Fitzgerald, J.E., Bito, H., *et al.* (2018). Long-term consolidation of ensemble neural plasticity patterns in hippocampal area CA1. *Cell Rep.* 25, 640-650 e642.
- Baddeley, A., Rubak, E., and Turner, R. (2015). Spatial point patterns: Methodology and applications with R, 1st edn (CRC Press).
- Bevins, R.A., and Besheer, J. (2006). Object recognition in rats and mice: a one-trial non-matching-to-sample learning task to study 'recognition memory'. *Nat. Protoc.* 1, 1306-1311.
- Davis, B.M., Rall, G.F., and Schnell, M.J. (2015). Everything you always wanted to know about rabies virus (but were afraid to ask). *Annu Rev. Virol.* 2, 451-471.
- Deng, W., Aimone, J.B., and Gage, F.H. (2010). New neurons and new memories: how does adult hippocampal neurogenesis affect learning and memory? *Nat Rev Neurosci* 11, 339-350.
- Deshpande, A., Bergami, M., Ghanem, A., Conzelmann, K.K., Lepier, A., Götz, M., *et al.* (2013). Retrograde monosynaptic tracing reveals the temporal evolution of inputs onto new neurons in the adult dentate gyrus and olfactory bulb. *Proc. Natl. Acad. Sci. USA* 110, E1152-1161.
- Friedman, A., Homma, D., Bloem, B., Gibb, L.G., Amemori, K.I., Hu, D., *et al.* (2017). Chronic stress alters striosome-circuit dynamics, leading to aberrant decision-making. *Cell* 171, 1191-1205 e1128.
- Gallitano, A.L., Satvat, E., Gil, M., and Marrone, D.F. (2016). Distinct dendritic morphology across the blades of the rodent dentate gyrus. *Synapse* 70, 277-282.
- Hooper, D.C., Roy, A., Barkhouse, D.A., Li, J., and Kean, R.B. (2011). Rabies virus clearance from the central nervous system. *Adv. Virus Res.* 79, 55-71.

Huang, L., Xi, Y., Peng, Y., Yang, Y., Huang, X., Fu, Y., *et al.* (2019). A visual circuit related to habenula underlies the antidepressive effects of light therapy. *Neuron* *102*, 128-142 e128.

Jahn, H.M., Kasakow, C.V., Helfer, A., Michely, J., Verkhatsky, A., Maurer, H.H., *et al.* (2018). Refined protocols of tamoxifen injection for inducible DNA recombination in mouse astroglia. *Sci. Rep.* *8*, 5913.

Kawashima, T., Kitamura, K., Suzuki, K., Nonaka, M., Kamijo, S., Takemoto-Kimura, S., *et al.* (2013). Functional labeling of neurons and their projections using the synthetic activity-dependent promoter E-SARE. *Nat. Methods* *10*, 889-895.

Lafon, M. (2011). Evasive strategies in rabies virus infection. *Adv. Virus Res.* *79*, 33-53.

Lavin, T.K., Jin, L., and Wickersham, I.R. (2019). Monosynaptic tracing: a step-by-step protocol. *J. Chem. Neuroanat.* *102*, 101661.

LeGates, T.A., Altimus, C.M., Wang, H., Lee, H.K., Yang, S., Zhao, H., *et al.* (2012). Aberrant light directly impairs mood and learning through melanopsin-expressing neurons. *Nature* *491*, 594-598.

Li, J.T., Xie, X.M., Yu, J.Y., Sun, Y.X., Liao, X.M., Wang, X.X., *et al.* (2017). Suppressed calbindin levels in hippocampal excitatory neurons mediate stress-induced memory loss. *Cell Rep.* *21*, 891-900.

McAvoy, K.M., Scobie, K.N., Berger, S., Russo, C., Guo, N., Decharatanachart, P., *et al.* (2016). Modulating neuronal competition dynamics in the dentate gyrus to rejuvenate aging memory circuits. *Neuron* *91*, 1356-1373.

Ming, G.L., and Song, H. (2005). Adult neurogenesis in the mammalian central nervous system. *Annu. Rev. Neurosci.* *28*, 223-250.

Mori, T., and Morimoto, K. (2014). Rabies virus glycoprotein variants display different patterns in rabies monosynaptic tracing. *Front. Neuroanat.* *7*, 47.

Nilssen, E.S., Doan, T.P., Nigro, M.J., Ohara, S., and Witter, M.P. (2019). Neurons and networks in the entorhinal cortex: a reappraisal of the lateral and medial entorhinal subdivisions mediating parallel cortical pathways. *Hippocampus* *29*, 1238-1254.

Osanai, Y., Shimizu, T., Mori, T., Yoshimura, Y., Hatanaka, N., Nambu, A., *et al.* (2017). Rabies virus-mediated oligodendrocyte labeling reveals a single oligodendrocyte myelinates

axons from distinct brain regions. *Glia* 65, 93-105.

Shansky, R.M., Hamo, C., Hof, P.R., McEwen, B.S., and Morrison, J.H. (2009). Stress-induced dendritic remodeling in the prefrontal cortex is circuit specific. *Cereb. Cortex* 19, 2479-2484.

Skelton, P.D., Frazel, P.W., Lee, D., Suh, H., and Luikart, B.W. (2019). Pten loss results in inappropriate excitatory connectivity. *Mol. Psychiatry* 24, 1627-1640.

Terreros-Roncal, J., Flor-García, M., Moreno-Jiménez, E.P., Pallas-Bazarra, N., Rábano, A., Sah, N., *et al.* (2019). Activity-dependent reconnection of adult-born dentate granule cells in a mouse model of frontotemporal dementia. *J. Neurosci.* 39, 5794-5815.

Turner, P.V., Brabb, T., Pekow, C., and Vasbinder, M.A. (2011). Administration of substances to laboratory animals: routes of administration and factors to consider. *J. Am. Assoc. Lab. Anim. Sci.* 50, 600-613.

Vallès, A., Boender, A.J., Gijssbers, S., Haast, R.A., Martens, G.J., and de Weerd, P. (2011). Genomewide analysis of rat barrel cortex reveals time- and layer-specific mRNA expression changes related to experience-dependent plasticity. *J. Neurosci.* 31, 6140-6158.

Wang, C., Yue, H., Hu, Z., Shen, Y., Ma, J., Li, J., *et al.* (2020). Microglia mediate forgetting via complement-dependent synaptic elimination. *Science* 367, 688-694.

Yang, G., Pan, F., and Gan, W.B. (2009). Stably maintained dendritic spines are associated with lifelong memories. *Nature* 462, 920-924.

Zhao, C., Teng, E.M., Summers, R.G., Jr., Ming, G.L., and Gage, F.H. (2006). Distinct morphological stages of dentate granule neuron maturation in the adult mouse hippocampus. *J. Neurosci.* 26, 3-11.

Zhou, W., Jin, Y., Meng, Q., Zhu, X., Bai, T., Tian, Y., *et al.* (2019). A neural circuit for comorbid depressive symptoms in chronic pain. *Nat. Neurosci.* 22, 1649-1658.

Zingg, B., Chou, X.L., Zhang, Z.G., Mesik, L., Liang, F., Tao, H.W., *et al.* (2017). AAV-mediated anterograde transsynaptic tagging: Mapping corticocollicular input-defined neural pathways for defense behaviors. *Neuron* 93, 33-47.

REVIEWER COMMENTS

Reviewer #1 (Remarks to the Author):

The manuscript from Wang and colleagues is a through, extensive and mutli-disciplinary study with novel findings regarding the impact of tactile experience on cognition. Furthermore, the authors have made significant efforts to address the reviewer's comments and should be commended.

The reviewer comments have been satisfied and the manuscript is recommended for publication.

Reviewer #2 (Remarks to the Author):

Wang et al., provide here a substantially revised version of their manuscript with additional experiments and figures to address the questions raised by thereferes. In particular, they performed new experiments to better characterise and describe the extent of neuronal activation in their novel tactile enrichment model (TEE). Additionally, the authors greatly improved the quality of the microscopy images. The authors have also addressed the lack of detailed information in some of the sections which produced confusion by providing more details. They provide explanations that help clarifying technical issues related to RABV-based connectivity tracing experiments. These are important points to help the reader to understand the rationale of the experimental scheme. As the fact that RABV remains active for long time is anything but obvious, this referee strongly suggests that the explanations provided in the rebuttal to the review are included in the main manuscript. The authors also tackle concerns raised in relation with the AAV-based anterograde tracing experiments. However, it is still not clear how the pattern observed in the AAV-based anterograde tracing experiments are produced: the authors did improve the quality of the pictures to render the axon-like structures more apparent. Yet, why do the authors see GFP+ somata in the LEC after injection of the anterograde tracer in S1? Is the AAV-virus capable of anterograde transsynaptic spread? If not, one would expect to see only axon-like structures in LEC, but no somata.

Finally, as a minor comment, the plots depicting the new nearest neighbor distance analysis are not very clear and maybe using another representation such as frequency distribution would be more appropriate.

Reviewer #3 (Remarks to the Author):

The authors have done an adequate job clarifying their analysis and interpretation, and there is now sufficient nuance in their conclusions. This study is suitable for publishing.

Title: Tactile Modulation of Memory and Anxiety Requires Dentate Granule Cells along the Dorsoventral Axis

Reviewer 2:

Wang et al., provide here a substantially revised version of their manuscript with additional experiments and figures to address the questions raised by the referees. In particular, they performed new experiments to better characterize and describe the extent of neuronal activation in their novel tactile enrichment model (TEE). Additionally, the authors greatly improved the quality of the microscopy images. The authors have also addressed the lack of detailed information in some of the sections which produced confusion by providing more details. They provide explanations that help clarifying technical issues related to RABV-based connectivity tracing experiments. These are important points to help the reader to understand the rationale of the experimental scheme. As the fact that RABV remains active for long time is anything but obvious, this referee strongly suggests that the explanations provided in the rebuttal to the review are included in the main manuscript.

Response:

Based on these constructive suggestions, we have cited relevant papers and integrated the rationale for our rabies virus delivery strategy, including the timing of rabies virus injection, the clearance of rabies virus in the central nervous system, and survival time between the first 4-hydroxytamoxifen injection and sacrifice, to the revised methods section (the last paragraph on page 27 and the first paragraph on page 28).

The authors also tackle concerns raised in relation with the AAV-based anterograde tracing experiments. However, it is still not clear how the pattern observed in the AAV-based anterograde tracing experiments are produced: the authors did improve the quality of the pictures to render the axon-like structures more apparent. Yet, why do the authors see GFP+ somata in the LEC after injection of the anterograde tracer in SI? Is the AAV-virus capable of anterograde transsynaptic spread? If not, one would expect to see only axon-like structures in LEC, but no somata.

Response:

The AAV2/1-Syn-Cre used in the current study could not only infect neurons near the injection site, but also infect postsynaptic cells that are targeted by these local neurons. As reported by Zingg et al. (<https://doi.org/10.1016/j.neuron.2016.11.045>), AAV serotype 1 has anterograde transsynaptic spread properties: after injection of AAV2/1-Syn-Cre to the primary visual cortex (V1) of Ai14 mice, which express tdTomato in a Cre recombinase-dependent manner, cells with soma and neurites labeled by tdTomato could be clearly seen in various regions directly innervated by V1. Consistent with their findings and recent studies, after delivery of this AAV to the S1 of Ai47 reporter mice, EGFP-expressing neurons in first-order downstream regions of S1, including LEC and some other regions not presented here (e.g., secondary somatosensory cortex and primary motor cortex), could be detected as expected, indicating that the AAV2/1-Syn-Cre we used did anterogradely spread from local S1 neurons to downstream structures, and expressed Cre and induced EGFP expression in infected S1-innervated cells. Additionally, we were able to label and manipulate S1-innervated LEC neurons by concomitant injections of this anterograde monosynaptic AAV to the S1 and the AAV-DIO-hM4Di-mCherry to the LEC. In the revised manuscript, we have specified the anterograde transsynaptic spread properties of this AAV to make the rationale clear (page 10, 2nd paragraph, line 4-6; page 28, 2nd paragraph, line 3).

Finally, as a minor comment, the plots depicting the new nearest neighbor distance analysis are not very clear and maybe using another representation such as frequency distribution would be more appropriate.

Response:

We thank the reviewer for this helpful suggestion. In the revised **Fig. 2d**, **Fig. 3g** and **Fig. 7e**, we have combined cumulative frequency plots with scatter plots, showing both data distribution and individual data points, to better present nearest neighbor distance results.

REVIEWERS' COMMENTS

Reviewer #2 (Remarks to the Author):

All remaining points were addressed adequately.